# A Systematic Review and Meta-Analysis of Ocular and Periocular Basal Cell Carcinoma with First-Time Description of Dermoscopic and Reflectance Confocal Microscopy Features of Caruncle Basal Cell Carcinoma

**DOI:** 10.3390/diagnostics15101244

**Published:** 2025-05-14

**Authors:** Martina Caviglia, Shaniko Kaleci, Pasquale Frascione, Miram Teoli, Maria Concetta Fargnoli, Giovanni Pellacani, Victor Desmond Mandel

**Affiliations:** 1Dermatologic Unit, Department of Clinical Internal, Anesthesiological and Cardiovascular Sciences, La Sapienza University of Rome, 00185 Rome, Italy; marticav8@gmail.com (M.C.); pellacani.giovanni@gmail.com (G.P.); 2Dermatology Unit, Surgical, Medical and Dental Department of Morphological Sciences Related to Transplant, Oncology and Regenerative Medicine, University of Modena and Reggio Emilia, 41124 Modena, Italy; shaniko_k@hotmail.com; 3Oncologic and Preventive Dermatology Unit, San Gallicano Dermatological Institute-IRCCS, 00144 Rome, Italy; pasquale.frascione@ifo.it; 4Porphyria and Rare Diseases Unit, San Gallicano Dermatological Institute-IRCCS, 00144 Rome, Italy; miriam.teoli@ifo.it; 5Department of Biotechnological and Applied Clinical Sciences, University of L’Aquila, 67100 L’Aquila, Italy; mariaconcetta.fargnoli@univaq.it

**Keywords:** neoplasms, tumor, carcinoma, basal cell, eye, ocular, periocular, eyelid, caruncle, dermoscopy, microscopy, confocal

## Abstract

**Background**: Basal cell carcinoma (BCC) of the ocular and periocular region is characterized by a painless progressive extension. An early diagnosis can limit the extent of facial tissue involvement and subsequent resection resulting in better cosmetic and functional results. **Objectives**: The aim is to provide the largest and most up-to-date overview of ocular and periocular BCCs. We also reported the first case of caruncle BCC investigated by dermoscopy and reflectance confocal microscopy (RCM). **Methods**: A systematic review and meta-analysis (Prospero ID CRD583032) were carried out by searching PUBMED–MEDLINE, including all articles with a full-text English version and with BCCs in eyelids, medial and lateral canthus, caruncle, conjunctiva, and orbit. The following data were collected: authors, year, title and type of publication, medical specialization, number, sex, age and comorbidities of the patients, anatomic localization of the disease, clinical and dermoscopic aspect, histological examination, and treatment. **Results**: We identified 731 articles through a database search, of which 236 articles matched our inclusion criteria. A total of 71.730 patients with ocular and periocular BCCs were included in the present study, and all data collected were reported in a dataset. Most of the articles included were described by ophthalmologists (67.5%), dermatologists (11.2%), or plastic surgeons (5.6%). The proportional meta-analysis revealed varying significance and heterogeneity for each type of study included. **Conclusions**: BCC more frequently affects the lower eyelid. The most common BCC subtype of ocular and periocular area is the nodular form. Limited data are available concerning the application of dermoscopy and RCM in this area. RCM may be particularly useful for early diagnosis, mapping, and treatment monitoring of ocular and periocular BCCs. Surgery still remains the first-choice treatment.

## 1. Introduction

Basal cell carcinoma (BCC) is a skin carcinoma that originates from epidermal cells. It is the most common malignant tumor in white populations, accounting for 75% of cases [1]. The incidence varies geographically; it also increases with age and is slightly more common in men. Several risk factors are involved in the pathogenesis, and ultraviolet radiation (UVR) is the most important cause [1]. BCC typically affects sun exposure areas, especially on the head and neck region, of which 20% occurs on the eyelids [2]. BCC is a malignant cancer that exceptionally metastasizes. The risk of recurrence may depend on the location of the tumor (H zone of the face), the histological characteristics, and immunosuppression [3]. The precise and early identification of these tumors can limit the extent of facial tissue involvement and subsequent resection resulting in better cosmetic and functional results. The diagnosis of BCC often requires a biopsy, especially in the case of ambiguous lesions. Dermoscopy usually allows for the early identification of BCC and preoperatively detects its subtype [1]. Additional non-invasive skin-imaging technology that has proven its high diagnostic value is in vivo reflectance confocal microscopy (RCM), which is often only accessible in specialized skin cancer centers [4].

RCM permits the capture of in vivo, cellular-resolution images of lesions, parallel to the skin surface, from the stratum corneum to the superficial dermis [5]. Acquisition of high-quality images with RCM can be impeded by technical difficulties in curved, or relatively inaccessible, surfaces such as ocular and periocular structures. Herein, we describe a rare case of caruncle BCC investigated with dermoscopy and RCM, after performing a systematic review and meta-analysis of all the available literature concerning ocular and periocular BCC. Currently, this is the first report on the application of non-invasive imaging tools in this anatomical location.

## 2. Materials and Methods

### 2.1. Search Strategy

A systematic review and meta-analysis of ocular and periocular BCCs reported in the literature were carried out by searching PUBMED–MEDLINE. Keywords used were “ocular basal cell carcinoma”, “periocular basal cell carcinoma”, “caruncle basal cell carcinoma”, “lacrimal caruncle basal cell carcinoma”, “ocular basal cell carcinoma AND confocal microscopy”, and “ocular basal cell carcinoma AND reflectance confocal microscopy”. PUBMED–MEDLINE was searched for studies published up to 31 December 2023.

### 2.2. Study Registration

This systematic review with meta-analysis was registered in PROSPERO (Prospero ID CRD583032) and conducted following PRISMA guidelines (Figure 1).

### 2.3. Data Collection

Data were independently extracted by two authors (MC and MT) and disagreement was resolved by a consensus or a third author (VDM) who acted as a referee. We included published articles with a full-text English version available and with specific BCCs’ localization in eyelids, medial and lateral canthus, caruncle, conjunctiva, and orbit. The search was restricted to studies on humans, while systematic and literature reviews were excluded. The following data were collected: authors, year, title and type of publication, medical specialization, number, sex, age and comorbidities of the patients, anatomic localization of the disease, clinical and dermoscopic aspect, histological examination, and treatment. Moreover, we reported the first case of caruncle BCC investigated with dermoscopy (VivaCam^®^: Mavig GmbH, Munich, Germany) and RCM (VivaScope^®^ 3000: Mavig GmbH, Munich, Germany). Instruments and the acquisition procedure have been described elsewhere [5,6].

### 2.4. Statistical Analysis

We performed a proportional meta-analysis using MedCalc 14.8.1 software (MedCalc Software bvba, Ostend, Belgium; http://www.medcalc.org; 2014, accessed on 1 September 2024) applying the Freeman–Tukey transformation (arcsine square root transformation) to calculate the weighted overall proportion (DerSimonian and Laird, 1986). Symptom proportions (expressed in percentage), within the 95% confidence interval (CI), from each study were included in the meta-analysis. The overall proportion within the 95% CI was calculated using both the random-effects model and the fixed-effects model. The fixed-effects model assumes that all included studies have a similar effect, so the summary effect is an estimate of the weight of similar effects in the studies. The random-effects model assumes that effects vary among studies, and the summary analysis is a weighted average reported in different studies.

A Forest plot was performed for each study included in the meta-analysis; values related to the effect size and CI were presented. The Forest plot also included the weighted effect of the prevalence of BCC by study type, with a 95% CI. The size of the marker (square) represents the weight of each study; studies with a smaller patient sample will have less weight. The overall effect is represented in the plot by a diamond: Its width represents precision, and its position represents the estimate of effects.

Heterogeneity among studies was estimated using Cochran’s Q statistic test and the I^2^ index. Heterogeneity, defined as whether observed variance exceeded expected variance, was considered significant when *p* < 0.01 for the Q statistic. The I^2^ index of given heterogeneity [I^2^ = 100% × (Q − df)/Q] was defined as I^2^ = 0–25%, homogeneous; I^2^ = 25–50%, moderate heterogeneity; I^2^ = 50–75%, large heterogeneity; I^2^ = 75–100%, extreme heterogeneity.

## 3. Results

### 3.1. Study Selection and Characteristics

We identified 731 articles through database search. Duplicates were omitted and 428 records were excluded (Appendix A) because they did not meet the inclusion criteria as mentioned above. A total of 236 articles were considered in our study (235 papers matched our criteria, while one was added after screening references of all selected studies) [7,8,9,10,11,12,13,14,15,16,17,18,19,20,21,22,23,24,25,26,27,28,29,30,31,32,33,34,35,36,37,38,39,40,41,42,43,44,45,46,47,48,49,50,51,52,53,54,55,56,57,58,59,60,61,62,63,64,65,66,67,68,69,70,71,72,73,74,75,76,77,78,79,80,81,82,83,84,85,86,87,88,89,90,91,92,93,94,95,96,97,98,99,100,101,102,103,104,105,106,107,108,109,110,111,112,113,114,115,116,117,118,119,120,121,122,123,124,125,126,127,128,129,130,131,132,133,134,135,136,137,138,139,140,141,142,143,144,145,146,147,148,149,150,151,152,153,154,155,156,157,158,159,160,161,162,163,164,165,166,167,168,169,170,171,172,173,174,175,176,177,178,179,180,181,182,183,184,185,186,187,188,189,190,191,192,193,194,195,196,197,198,199,200,201,202,203,204,205,206,207,208,209,210,211,212,213,214,215,216,217,218,219,220,221,222,223,224,225,226,227,228,229,230,231,232,233,234,235,236,237,238,239,240,241,242]. All data collected are reported in a dataset (Table 1), while the characteristics of the studies are summarized in Table 2. The articles selected were mainly retrospective (32.6%), case series (32.6%), and reports (22.8%). Most of the articles included were described by ophthalmologists (67.5%), dermatologists (11.2%), and plastic surgeons (5.6%) but also less frequently by other specialists. A total of 71.730 patients with ocular and periocular BCCs are included in the present systematic review with meta-analysis and their characteristics are described in Table 3. Different anatomical locations are summarized in Table 4. “Eyelid and periocular area” (48.7%) and “eyelid” (29.9%) were the two more commonly involved sites, while “periocular”, “ocular”, and “caruncle” areas were only rarely interested. Clinic, dermoscopy, RCM, and histopathology data are included in Table 5, while treatments are given in Table 6.

### 3.2. Case

In September 2022, a 38-year-old lady was referred to San Gallicano Dermatological Institute-IRCCS, for evaluation of a caruncle pigmented lesion on the left eye. Her family and medical history were unremarkable. The lesion, which appeared approximately 7 months ago, was indolent. Clinical examination revealed a millimetric flat lesion, with faded and irregular margins, black-blue in color (Figure 2a). After applying oil immersion fluid, dermoscopic examination with VivaCam^®^ showed the presence of blue–black structureless areas (Figure 2b,c). Sharply demarcated lobular nests, outlined by dark peritumoral clefting, were detected with the RCM handheld probe and corresponded to basaloid islands (Figure 2). Polarized elongated keratinocytes (streaming) characterized the overlying skin (Figure 2d). Hyper-refractile thin dendrites and bright oval structures were observed within and around the tumor island (Figure 2e,f), corresponding to melanophages, while peripheral palisading of nuclei can also be detected at higher magnification. The adjacent dermal stroma contained small bright dots compatible with smaller inflammatory cells. Convoluted and dilated blood vessels coursing in the horizontal plane of imaging were seen in real-time or video-mode RCM imaging, juxtaposed to the tumor islands (Appendix A). Based on RCM findings, a diagnosis of pigmented BCC was made, and the lesion was subsequently surgically removed by an ophthalmologist. Histopathologic examination confirmed the diagnostic suspicion and the tumor was completely excised with a 2 mm of free margins. The patient did not complain of any discomfort or functional impairment. At a two-year follow-up, no signs of local recurrence were observed.

### 3.3. Results of Meta-Analysis

The results of the proportional meta-analysis, including the combined proportion (95% CI), are summarized in Table 7, with estimates of the overall proportion shown in the Forest Plot (Figure 3, Figure 4, Figure 5, Figure 6, Figure 7 and Figure 8). The overall proportion for “clinical trial” was 97.6% (91.6–99.9), demonstrating significant and extreme heterogeneity (Q = 122.7, df = 3, I^2^ = 75.6%, *p* = 0.006). For “prospective study” and “prospective case series”, the overall proportion was 93.0% (79.2–99.6), indicating significant and extreme heterogeneity (Q = 40.6, df = 20, I^2^ = 99.5%, *p* < 0.001). “Retrospective study” showed an overall proportion of 61.1% (48.8–72.8), with significant and extreme heterogeneity (Q = 346.3, df = 75, I^2^ = 99.8%, *p* < 0.001). “Case reports” and “case report with review of the literature” studies showed an overall proportion of 85.3% (77.7–91.0), with non-significant and homogeneous (Q = 77.8, df = 53, I^2^ = 0.0%, *p* = 1.000). “Case series” studies showed an overall proportion of 73.3% (62.1–83.3), with significant and extreme heterogeneity (Q = 199.1, df = 66, I^2^ = 99.6%, *p* < 0.001) and “letter” showed an overall proportion of 89.9% (72.8 to 99.0), with significant and moderate heterogeneity (Q = 233.3, df = 8, I^2^ = 65.7%, *p* = 0.003).

## 4. Discussion

### 4.1. Epidemiology and Localization

In the development of BCC, the main risk factors include UV radiation exposure, fair pigmentary characteristics, older age, genodermatoses, a family and personal history of BCC, and immunosuppression [1]. UVB radiation (290–320 nm) is believed to play a greater role than UVA radiation (320–400 nm). Different carcinogens can target different stem cell compartments and subsequently give rise to BCCs. Several hypotheses have been formulated regarding the origin of BCCs. Most BCCs seem to derive from hair follicle stem cells, while some authors claim that BCC stem cells are in the interfollicular epidermis. There is generally a latency period of several years between sun damage and the onset of BCC. Thereby, BCC develops in chronically photo-exposed areas most commonly in the head and neck [243] and accounts for 90% of malignant eyelid tumors [2]. Due to changes in sun exposure behavior and the general aging of the population, an increase in the incidence of BCC has been observed in many countries around the world. Wide regional variations in the reported incidence rates of BCC have been found, due to the latitude of the population, the study period, and the methods of recording BCC [244]. Generally, BCC arises in the elderly population. We observed an age range of 9 to 105 years for all cases examined. In 45 reports, the age range is not available. However, some patients can develop this skin cancer at an earlier age (<40 years), and patients with genetic syndromes such as Xeroderma Pigmentosum (XP) or Basal Cell Naevus Syndrome (BCNS or Gorlin–Goltz) can develop BCC earlier, even before 20 years of age [3]. In our study, 21 patients were affected by BCNS with a median age of 43 years and 27 patients suffered from XP with a mean age of 21 years. An increasing trend has been observed for patients aged ≥ 50 years over the next 10 years, and the incidence of BCC is expected to increase by 30–40% (males) and 25–30% (females) [244]. BCC is a tumor that affects both sexes, and no sex predilection has been observed [245]. In the literature, some authors, such as Saleh GM et al. in 2017 [246] in an article on eyelid basal cell carcinoma in England, reported a slight preponderance in males. Dessinioti C et al., in 2010 [247], reported a female-to-male ratio of 2:1. Our analysis showed a mild prevalence of the female sex (53.4%). However, in 91 articles, there were no data regarding the sex of patients. In the case we described, BCC occurred in a woman, who was younger than the average age reported in the literature even with a negative medical history.

The site most frequently affected by BCC is the lower eyelid, followed by the medial canthus, upper eyelid, and lateral canthus [49,63,248]. The involvement of the lower eyelid could be the consequence of light reflection by the cornea, and other irritant chemical or physical insults could be considered [109,249,250,251,252]. The uncommon envelopment of the upper eyelid could be due to the protection of the eyebrow [251].

Our analysis showed that in 193 (81.8%) articles, there was eyelid involvement; in 138 (58.5%), the upper/lower eyelid location was specified; among these, the involvement of both was reported in 89 papers (37.7%); in 38 (16.1%), just the lower eyelid was involved and in 11 (4.7%), just the upper eyelid. In the ocular and periocular regions, the growth of the tumors is characterized by a painless progressive extension; therefore, we often have the involvement of contiguous ocular structures [249].

On the eyelids, the BCC rapidly invades the dermis, followed by the infraorbital extension. On the medial canthus, the tumor can spread on the orbit and then can destroy the globe. Primary BCCs of the mucocutaneous transition region or the conjunctiva are extremely rare [252]. These sites are more frequently affected secondarily by infiltration from the tumor of the eyelid or canthus region [52]. Malignancies of the caruncle are generally very rare with a frequency of 5% among all the caruncle tumors [253,254,255]. In addition to our case of primary caruncle BCC, we identified 11 articles in the literature describing caruncle BCCs in a total of 15 patients with a median age of 61.1 years (range 24–84 years) [12,25,30,51,59,61,62,76,101,113,153].

### 4.2. Diagnosis

BCCs show polymorphic clinical aspects and different dermoscopical features due to their anatomical location [256,257,258]. The nodular BCC subtype occurs predominantly on the head–neck region and accounts for 60% of all BCCs. The superficial subtype represents 20% of BCCs and mainly involves the trunk, while other subtypes are less frequent. The clinical aspect of BCCs was reported in only 68 articles (28.8%). Palpable lesions resulted in the most common form (73.5%) and were described as “tumors”, “masses”, “papular”, “nodular”, or “exophytic lesions”, followed by ulcerated (36.8%) and pigmented (17.6%) lesions. Instead, sclerosing morphic-like lesions were reported in only two articles. Regarding primary caruncle BCCs, most of the papers (72.7%) mentioned the clinical aspect and described the nodular (33.3%) and pigmented (20%) lesions as the more common type. Dermatoscopic features were reported in only three articles (none regarding caruncle) of our systematic review with meta-analysis. The cause may be associated with the difficulty of exploring this anatomical area with dermatoscopy instruments. However, it is noted that only 11% of the papers were published by dermatologists, while the majority concerned ophthalmologists (67.5%). Non-dermatological assessment is probably the other main reason for the scarcity of dermatoscopic data in literature. Clinical diagnosis confirmed on dermatoscopy without histopathological examination is considered acceptable only for the small nodular form on typical locations such as the head/neck or trunk, and for the superficial BCC located on the trunk and extremities [257]. Classification of BCCs into low and high risks is based on the probability of recurrence. This risk is related to the localization on the H area of the face, aggressive histological features (perineural and/or vascular involvement), and/or immunosuppression. In case of doubtful lesions as well as high-risk BCCs, histological correlation is mandatory [256,257]. In low-risk BCCs, imaging techniques may be sufficient to confirm the diagnostic suspicion [257]. RCM has been arousing great interest in recent years in the diagnosis of skin tumors and particularly, the diagnosis of eyelid margin lesions to minimize the number of unnecessary surgical excisions [123]. Identification of RCM criteria is particularly important in identifying BCC whose clinical and dermoscopical appearance may mimic other malignant or benign lesions [259]. The RCM handheld probe can be applied directly to the skin, and it allows imaging of lesions in less accessible sites, such as the structure of the eye and oral and genital mucosa [260,261,262,263]. Currently, there is only one study published regarding the application of RCM in the eyelid area [123]. Among 47 eyelid margin lesions, the diagnosis of BCC was made in 14 cases and based on the recognition of at least 2 of the following criteria: dark silhouette, lobular nests or trabecular structures of tightly packed cells, peripheral palisading of elongated cells, peritumoral clefts, convoluted and dilated blood vessels, and polarized elongated keratinocytes (streaming) of the overlying skin. RCM sensitivity and specificity in this study resulted in 100% and 60%, respectively. In our caruncle BCC, we observed with RCM the presence of streaming, lobular nests with peripheral palisading, peritumoral clefts, and dilated blood vessels juxtaposed to the tumor islands, confirming the utility of the criteria identified by Cinotti et al. [123]. In our study, histological diagnosis was reported in 234 articles, and among them, the subtype was specified in 40.7% (nodular in 71% of the papers).

Multiple recurrences are linked to aggressive subtype, male gender, large lesion size, perineural invasion, medial canthal localization, and advanced patient age [90,264,265]. A study on 63 patients revealed that medial canthus lesions appear to have a higher risk of orbital invasion in comparison to the other ocular and periocular areas [47]. The cause may be associated with the difficulty of performing a complete surgical excision in this anatomical zone [266,267].

### 4.3. Treatment

BCCs belong to a special class of tumors characterized by a slow, persistent, and locally invasive growth pattern. If inadequately treated, it may progress into a large and deeply infiltrating locally advanced tumor (laBCC) or rarely (from 0.0028% to 0.55%) in a metastatic BCC (mBCC) [268].

A recent EADO classification introduced the concept of “easy to treat” and “difficult to treat” BCCs [257]. “Difficult to treat” BCCs included mBCC, laBCCs, and other types which, for any reason, pose specific management difficulties. For the treatment of BCC, different modalities can be used, but only surgical excision provides histological confirmation of successful treatment [269]. Moreover, surgeries still have the lowest recurrence rates, but they can cause functional loss or cosmetic disfigurement and have a risk of bleeding and infections [270]. Topical therapy and destructive approaches may be considered in patients with superficial BCC, while photodynamic therapy (PDT) can be an option for superficial and low-risk nodular BCCs [257]. Moreover, combined modalities such as laser CO_2_ and PDT could be used for selected patients with problematic reconstruction after surgery, aesthetic reasons, for patients with numerous and frequently occurring BCCs like patients affected by Gorlin–Goltz syndrome or patients who have undergone transplantation, for patients for whom anesthesia may be problematic, for patients receiving oral anticoagulants, and finally for patients who refused other treatments [269,270]. Regarding laBCCs and mBCC, treatments included hedgehog inhibitors, immunotherapy with anti-PD1, radiotherapy, and electrochemotherapy [257].

In our study, the only use of surgical techniques is reported in most of the articles (65.3%), confirming to be the first-choice treatment. Combined modality (surgery plus other treatments) found their application in 27.1% of papers, followed by non-surgical therapies in 6.8%. Only two articles did not mention the therapeutic approach. In our case of caruncle BCC, the therapeutic choice was standard surgical excision with 2 mm of free margins. We highlight the importance of RCM for the diagnosis and treatment monitoring of BCC because it allows to control histologic clearance and detect early recurrences [269,270]. Furthermore, RCM facilitates the presurgical and intrasurgical lateral and deep margin assessment of poorly defined BCCs, especially on aesthetically relevant sites such as ocular and periocular areas [271].

## 5. Conclusions

Our systematic review and meta-analysis collected the largest and most up-to-date collection of ocular and periocular BCCs. The site most frequently affected by BCC is the lower eyelid, followed by medial canthus which has a higher risk of orbital invasion, while primary caruncle BCC is extremely rare. Nodular BCC is the most common subtype of ocular and periocular areas. Dermoscopic and RCM studies concerning these areas are few in the literature. RCM may be very useful for early diagnosis, mapping, and treatment monitoring, especially in aesthetically relevant sites. Surgery still remains the first-choice treatment for ocular and periocular BCC, even if studies regarding the use of combined modality are increasing over time.

## Figures and Tables

**Figure 1 diagnostics-15-01244-f001:**
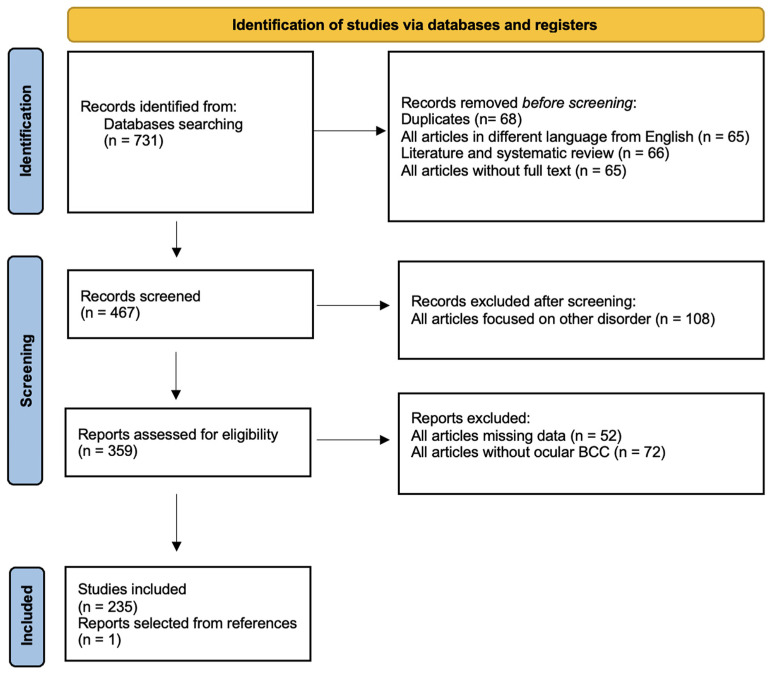
Flow diagram of the study selection process.

**Figure 2 diagnostics-15-01244-f002:**
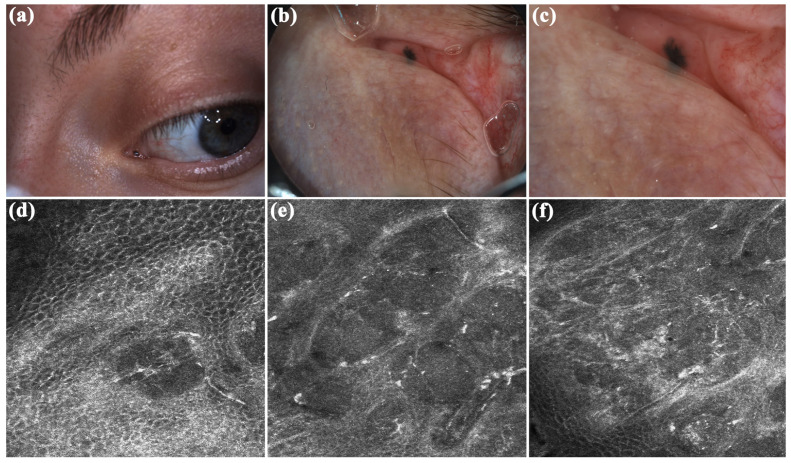
Caruncle basal cell carcinoma. (**a**) Clinically appeared as a millimetric flat lesion, with faded and irregular margins, blue–black in color. (**b**,**c**) Dermoscopic assessment revealed black–blue structureless areas. (**d**–**f**) Reflectance confocal microscopy (RCM) showed the presence of sharply demarcated lobular nests, outlined by dark peritumoral clefting. Streaming, peripheral palisading, hyper-refractile thin dendrites and bright oval structures, and convoluted and dilated blood vessels are other RCM criteria detected.

**Figure 3 diagnostics-15-01244-f003:**
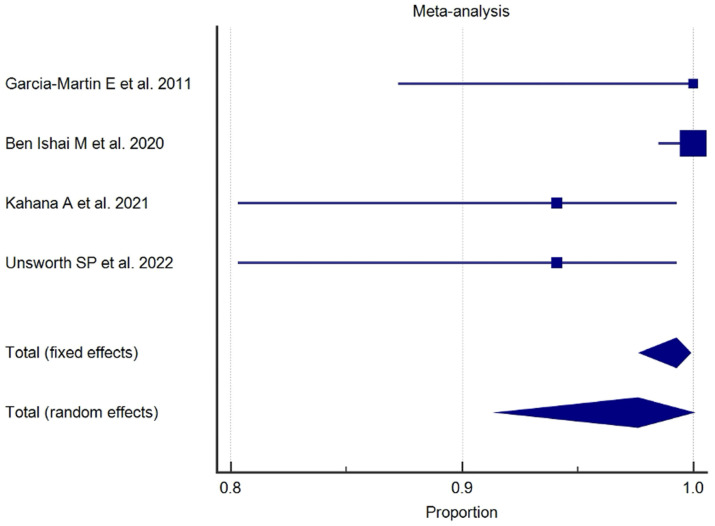
Proportional meta-analysis of the included “clinical trial” studies.

**Figure 4 diagnostics-15-01244-f004:**
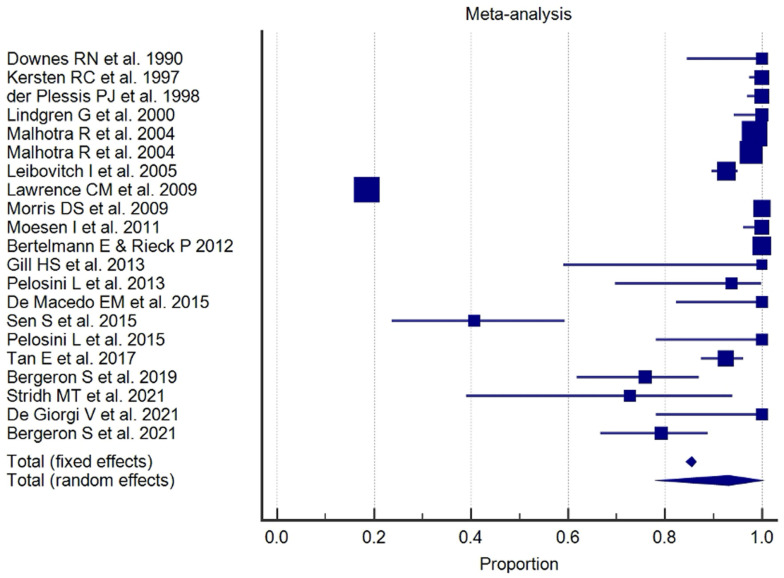
Proportional meta-analysis of the included “prospective study” and “prospective case series”.

**Figure 5 diagnostics-15-01244-f005:**
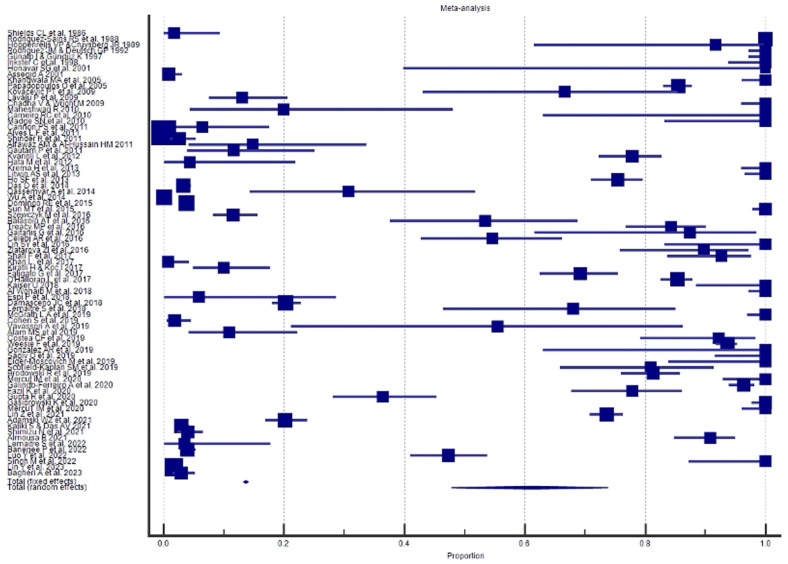
Proportional meta-analysis of the included “retrospective study”.

**Figure 6 diagnostics-15-01244-f006:**
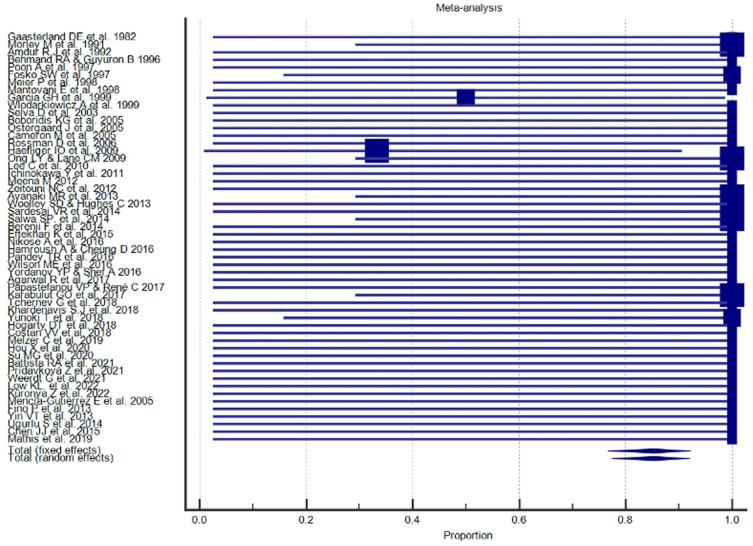
Proportional meta-analysis of the included “case reports” and “case report with review of the literature”.

**Figure 7 diagnostics-15-01244-f007:**
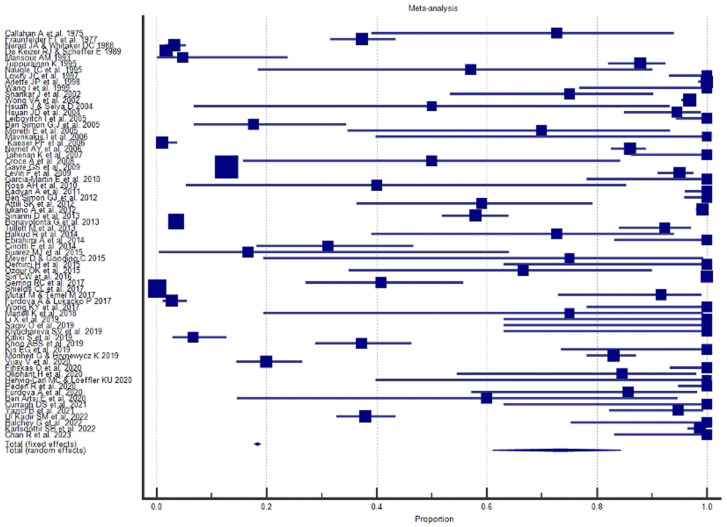
Proportional meta-analysis of the included “case series”.

**Figure 8 diagnostics-15-01244-f008:**
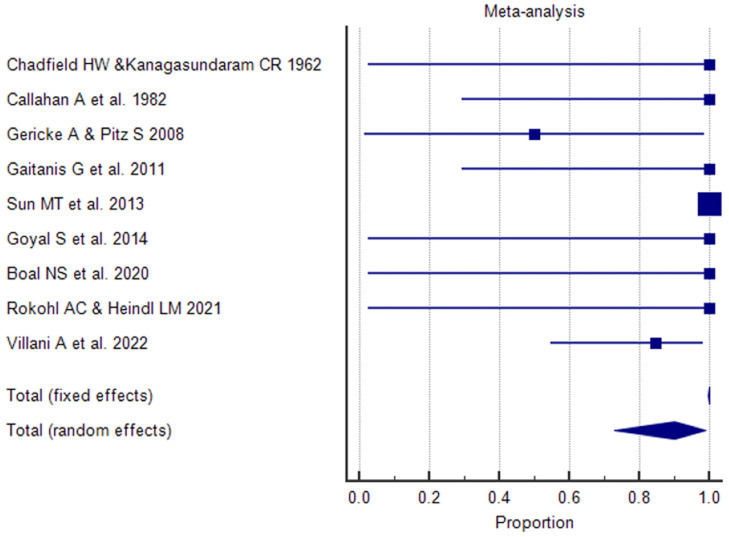
Proportional meta-analysis of the included “letter”.

**Table 1 diagnostics-15-01244-t001:** Dataset.

Authors (Year) [Reference]	Title	Type of Article	Specialization	Cases	Gender (n° pz)	Age (Mean/ Median Age)	Comorbidities (n° pz)	Anatomic Site	Anatomic Area	Clinical Aspect	Dermoscopy	Histological Examination (n° pz)	Treatment
Chadfield HW and Kanagasundaram CR (1962) [7]	Carcinoma in benign mucous membrane pemphigoid (ocular pemphigus)	Letter	Dermatology	1	M (1)	56	Ocular pemphigus (1)	Eyelid Medial canthus	Eyelid Periocular area	Ulcerated nodule	NP	Solid BCC (1)	Surgery + RT
Callahan A et al. (1975) [8]	Massive orbital invasion by small malignant lesions	Case series	Ophthalmology	8 (among 11 cases)	M (6) F (2)	Range 43–71 (55.8/55)	NA	Upper eyelid Lower eyelid Medial canthus Lateral canthus	Eyelid Caruncle Periocular area	Ulcerated lesion	NP	Nodular BCC (1) Morpheaform BCC (3) Ulcerative and invasive BCC (4)	Surgery ± RT
Fraunfelder FT et al. (1977) [9]	Cryosurgery for ocular and periocular lesions	Case series	Ophthalmology	101 (among 270 cases)	NA	NA	NA	Eyelid Medial canthus Lateral canthus Lacrimal outflow	Eyelid Ocular area Periocular area	NA	NP	BCC (101)	Cryosurgery
Gaasterland DE et al. (1982) [10]	Ocular involvement in xeroderma pigmentosum	Case reports	Ophthalmology	1	F (1)	36	Xeroderma pigmentosum (1)	Lower eyelid	Eyelid	Nodule	NP	BCC (1)	Cryotherapy + Surgery
Callahan A et al. (1982) [11]	Cancer excision from eyelids and ocular adnexa: the Mohs fresh tissue technique and reconstruction	Letter	Ophthalmology	3	M (3)	Range 48–80 (63/61)	NA	Lower eyelid Medial canthus Lateral canthus	Eyelid Periocular area	NA	NP	BCC (3)	Surgery ± RT
Shields CL et al. (1986) [12]	Types and frequency of lesions of the caruncle	Retrospective study	Pathology	1 (among 57 cases)	F (1)	61	NA	Caruncle	Caruncle	NA	NP	BCC (1)	Surgery
Nerad JA and Whitaker DC (1988) [13]	Periocular basal cell carcinoma in adults 35 years of age and younger	Case series	Ophthalmology	13 (among 409 cases)	M (6) F (7)	Range 21–35 (28.9/29)	Basal cell nevus syndrome (3)	Upper eyelid Lower eyelid Medial canthus Lateral canthus	Eyelid Periocular area	Erythematous, flat or slightly raised lesion	NP	Morpheaform BCC (7) Nodular BCC (6)	Surgery + Cryotherapy
Rodriguez-Sains RS et al. (1988) [14]	Radiotherapy of periocular basal cell carcinomas: recurrence rates and treatment with special attention to the medical canthus	Retrospective study	Ophthalmology	631	NA	NA	NA	Eyelid	Eyelid	NA	NP	BCC (631)	Surgery ± RT or other therapy
Hoppenreijs VP and Cruysberg JR (1989) [15]	Spontaneous repair of lower eyelid after tumour excision	Retrospective study	Ophthalmology	11 (among 12 cases)	M (5) F (6)	Range 48–77 (66.1/69)	NA	Lower eyelid	Eyelid	NA	NP	BCC (11)	Surgery
De Keizer RJ and Scheffer E (1989) [16]	Masquerade of eyelid tumours	Case series	Ophthalmology	15 (among 830 cases)	NA	NA	NA	Eyelid	Eyelid	NA	NP	BCC (15)	Surgery
Downes RN et al. (1990) [17]	Micrographic (MOHS’) surgery in the management of periocular basal cell epitheliomas	Prospective case series	Ophthalmology	22	M (8) F (14)	Range 27–85 (67/NA)	NA	Upper eyelid Lower eyelid Medial canthus Lateral canthus	Eyelid Periocular area	NA	NP	BCC (22)	Surgery + RT
Morley M et al. (1991) [18]	Cis-platinum chemotherapy for ocular basal cell carcinoma	Case reports	Ophthalmology	3	M (2) F (1)	Range 66–80 (71.3/68)	Mental confusion (1)	Lower eyelid	Eyelid	Nodule	NP	BCC (2) Infiltrating morpheaform BCC (1)	Surgery + Cis-platinum chemotherapy + Surgery or EBRT
Rodriguez JM and Deutsch GP (1992) [19]	The treatment of periocular basal cell carcinomas by radiotherapy	Retrospective study	Radiotherapy	127	NA	Range 40–94 (62/NA)	NA	Upper eyelid Lower eyelid Medial canthus Lateral canthus	Eyelid Periocular area	NA	NP	BCC (127)	RT + Surgery
Amdur R.J et al. (1992) [20]	Radiation therapy for skin cancer near the eye: kilovoltage X-rays versus electrons	Case reports	Radiotherapy	1	NA	NA	NA	Lower eyelid	Eyelid	NA	NP	BCC (1)	RT
Mansour AM (1993) [21]	Adnexal findings in AIDS	Case series	Ophthalmology	1 (among 21 cases)	M (1)	31	AIDS and pneumocystis carinii pneumonia (PCP) (1)	Lower eyelid	Eyelid	NA	NP	BCC (1)	Surgery
Tuppurainen K (1995) [22]	Cryotherapy for eyelid and periocular basal cell carcinomas: outcome in 166 cases over an 8-year period	Case series	Ophthalmology	146 (among 166 cases)	NA	Range 30–89 (67.9/NA)	NA	Upper eyelid Lower eyelid Medial canthus Lateral canthus	Eyelid Periocular area	NA	NP	BCC (142) Morpheaform BCC (4)	Surgery + Cryotherapy
Naugle TC et al. (1995) [23]	Free graft enhancement using orbicularis muscle mobilization	Case series	Ophthalmology	4 (among 7 cases)	M (1) F (3)	Range 48–83 (67.5/69)	NA	Upper eyelid Lower eyelid Medial canthus Lateral canthus	Eyelid Periocular area	NA	NP	BCC (4)	Surgery
Behmand RA and Guyuron B (1996) [24]	Resection of bilateral orbital and cranial base basal cell carcinoma with preservation of vision	Case reports	Plastic surgery	1	F (1)	68	NA	Both orbits	Ocular area	Ulcerated nodule	NP	BCC (1)	Surgery
Poon A et al. (1997) [25]	Primary basal cell carcinoma of the caruncle	Case reports	Ophthalmology	1	M (1)	74	SCC, BCC, and glaucoma (1)	Caruncle	Caruncle	Vascular, pink nodule	NP	Solid-microcystic BCC (1)	Surgery
Günalp I and Gündüz K (1997) [26]	Secondary orbital tumors	Retrospective study	Ophthalmology	129 (among 524 cases)	NA	Range 37–84 (58.4/NA)	NA	Eyelid	Eyelid	NA	NP	BCC (129)	Surgery
Fosko SW et al. (1997) [27]	Basal cell carcinoma involving the lacrimal canaliculus. A documented mechanism of tumor spread	Case reports	Dermatology	2	M (1) F (1)	30, 62	NA	Lacrimal canaliculus	Periocular area	Papule	NP	BCC (2)	Surgery
Lowry JC et al. (1997) [28]	The role of second-intention healing in periocular reconstruction	Case series	Ophthalmology	52 (among 65 cases)	M (26) F (26)	Range 44–85 (68.7/71)	NA	Upper eyelid Lower eyelid Medial canthus	Eyelid Periocular area	NA	NP	BCC (52)	Surgery
Kersten RC et al. (1997) [29]	Accuracy of clinical diagnosis of cutaneous eyelid lesions	Prospective study	Ophthalmology	139 (among 864 cases)	NA	Range 31–96 (72/NA)	NA	Upper eyelid Lower eyelid Medial canthus Lateral canthus	Eyelid Periocular area	NA	NP	BCC (139)	Surgery
Meier P et al. (1998) [30]	Primary basal cell carcinoma of the caruncle	Case reports	Ophthalmology	1	M (1)	24	NA	Caruncle	Caruncle	Whitish nodule with a reddish center and fine vessels	NP	Solid-cystic BCC (1)	Surgery
Mantovani E et al. (1998) [31]	Recurrent eyelid basal cell carcinoma with sclerochoroidal infiltration: echographic findings	Case reports	Ophthalmology	1	F (1)	82	NA	Upper eyelid	Eyelid	NA	NP	BCC (1)	Surgery
Arlette JP et al. (1998) [32]	Basal cell carcinoma of the periocular region	Case series	Dermatology	233	M (106) F (127)	Range 17–102 (58.8/59)	NA	Upper eyelid Lower eyelid Medial canthus Lateral canthus	Eyelid Periocular area	NA	NP	BCC (233)	Surgery ± Electrodesiccation and curettage or cryotherapy or RT
der Plessis PJ et al. (1998) [33]	Mohs’ surgery of periocular basal cell carcinoma using formalin-fixed sections and delayed closure	Prospective study	Dermatology	120	M (61) F (59)	Range 25–94 (NA/68)	NA	Upper eyelid Lower eyelid Medial canthus	Eyelid Periocular area	NA	NP	BCC (120)	Surgery ± RT ± Curettage and cautery or cryotherapy
Inkster C et al. (1998) [34]	Oculoplastic reconstruction following Mohs surgery	Retrospective study	Ophthalmology	57 (among 60 cases)	NA	Range 34–91 (68/NA)	NA	Upper eyelid Lower eyelid Medial canthus Lateral canthus	Eyelid Periocular area	NA	NP	BCC (57)	Surgery
Garcia GH et al. (1999) [35]	Periocular deep cutaneous basal cell carcinoma	Case reports	Ophthalmology	1 (among 2 cases)	F (1)	36	NA	Medial canthus	Periocular area	Nodule	NP	BCC (1)	Surgery
Wlodarkiewicz A et al. (1999) [36]	Extensive periocular defect reconstruction with local flaps and conchal cartilage graft	Case reports	Dermatology	1	F (1)	77	NA	Lateral canthus	Periocular area	Ulcerated nodule	NP	Infiltrative BCC (1)	Surgery
Carter KD et al. (1999) [37]	Clinical factors influencing periocular surgical defects after Mohs micrographic surgery	Comparative study	Ophthalmology	250 (among 281 cases)	NA	Range 14–100 (67.5/71)	Bazek syndrome (1); Basal cell nevus syndrome (3)	Upper eyelid Lower eyelid Medial canthus Lateral canthus	Eyelid Periocular area	NA	NP	Nodular/nodular ulcerative BCC (230) Morpheaform BCC (18) Multicentric BCC (2)	Surgery
Wang I et al. (1999) [38]	Photodynamic therapy utilising topical delta-aminolevulinic acid in non-melanoma skin malignancies of the eyelid and the periocular skin	Case series	Oncology	14	M (6) F (8)	Range 38–89 (77/79.5)	Basal cell nevus syndrome (1)	Upper eyelid Lower eyelid Medial canthus Lateral canthus	Eyelid Periocular area	Nodule	NP	Nodular cystic BCC (6) Nodular/morpheaform BCC (2) Nodular ulcerative (6)	PDT
Lindgren G et al. (2000) [39]	Mohs’ micrographic surgery for basal cell carcinomas on the eyelids and medial canthal area. II. Reconstruction and follow-up	Prospective case series	Ophthalmology	61	M (19) F (42)	Range 31–92 (73/NA)	NA	Upper eyelid Lower eyelid Medial canthus Lateral canthus	Eyelid Periocular area	NA	NP	Superficial BCC (NA) Micronodular (NA) Morpheaform BCC (NA)	Surgery
Honavar SG et al. (2001) [40]	Basal cell carcinoma of the eyelid associated with Gorlin-Goltz syndrome	Retrospective study	Ophthalmology	4	M (1) F (3)	Range 67–86 (72.8/69)	Basal cell nevus syndrome (4)	Upper eyelid Lower eyelid Medial canthus	Eyelid Periocular area	Ulcerated nodule	NP	BCC (4)	Surgery + EBRT
Rohrbach JM (2001) [41]	Immunology and growth characteristics of ocular basal cell carcinoma	Comparative study	Ophthalmology	20	M (10) F (10)	Range 41–90 (NA/77)	NA	Eyelid Medial canthus	Eyelid Periocular area	NA	NP	Nodular BCC (17) Fibrosing BCC (3)	Surgery
Assegid A (2001) [42]	Pattern of ophthalmic lesions at two histopathology centres in Ethiopia	Retrospective study	Ophthalmology	2 (among 231 cases)	NA	Range 15–>60 (NA/NA)	NA	Eyelid	Eyelid	NA	NP	BCC (2)	Surgery
Shankar J et al. (2002) [43]	Management of peri-ocular skin tumours by laissez-faire technique: analysis of functional and cosmetic results	Case series	Ophthalmology	18 (among 24 cases)	NA	Range 32–94 (66/NA)	NA	Upper eyelid Lower eyelid Medial canthus Lateral canthus	Eyelid Periocular area	NA	NP	BCC (18)	Surgery
Wong VA et al. (2002) [44]	Management of periocular basal cell carcinoma with modified en face frozen section controlled excision	Case series	Ophthalmology	633 (among 653 cases)	NA	Range NA (64/NA)	NA	Upper eyelid Lower eyelid Medial canthus Lateral canthus	Eyelid Periocular area	NA	NP	Solid BCC (450) Infiltrating BCC (123) Superficial/multifocal BCC (80) Micronodular BCC (47) Morpheic BCC (35) Adenoid BCC (15) Pigmented BCC (6) Unspecified/unavailable (57)	Surgery
Selva D et al. (2003) [45]	Recurrent morphoeic basal cell carcinoma at the lateral canthus with orbitocranial invasion	Case reports	Ophthalmology	1	F (1)	85	No comorbidities	Lateral canthus	Periocular area	Plaque	NP	Nodulocystic/morpheic BCC (1)	Surgery + RT
Hsuan J and Selva D (2004) [46]	Early division of a modified Cutler-Beard flap with a free tarsal graft	Case series	Ophthalmology	2 (among 4 cases)	M (1) F (1)	69, 78	NA	Upper eyelid	Eyelid	NA	NP	BCC (2)	Surgery
Malhotra R et al. (2004) [47]	The Australian Mohs database, part I: periocular basal cell carcinoma experience over 7 years	Prospective case series	Ophthalmology	1.275 (among 1.295 cases)	NA	Range 15–92 (60/61)	NA	Upper eyelid Lower eyelid Medial canthus	Eyelid Periocular area	NA	NP	Infiltrating BCC (431) Micronodular BCC (63) Nodulocystic BCC (489) Morpheaform BCC (51) Superficial BCC (188)	Surgery
Hsuan JD et al. (2004) [48]	Small margin excision of periocular basal cell carcinoma: 5 year results	Case series	Ophthalmology	52 (among 55 cases)	NA	Range 34–91 (66/NA)	NA	Upper eyelid Lower eyelid Medial canthus Lateral canthus	Eyelid Periocular area	NA	NP	Nodular BCC (52)	Surgery
Malhotra R et al. (2004) [49]	The Australian Mohs database, part II: periocular basal cell carcinoma outcome at 5-year follow-up	Prospective case series	Ophthalmology	801 (among 819 cases)	NA	NA	NA	Upper eyelid Lower eyelid Medial canthus	Eyelid Periocular area	NA	NP	Infiltrating BCC (234) Micronodular BCC (41) Nodulocystic BCC (338) Morpheaform BCC (35) Superficial BCC (107) Combined nodular and superficial BCC (1) Combined superficial and micronodular (1)	Surgery
Boboridis KG et al. (2005) [50]	Combination of periocular myocutaneous flaps for one-stage reconstruction of extensive defects of the eyelid	Case reports	Ophthalmology	1	M (1)	76	Renal failure, hypertension, and glaucoma (1)	Lower eyelid	Eyelid	Eroded nodule	NP	Nodular BCC (1)	Surgery
Ostergaard J et al. (2005) [51]	Primary basal cell carcinoma of the caruncle with seeding to the conjunctiva	Case reports	Ophthalmology	1	F (1)	60	No comorbidities	Caruncle	Caruncle	Pale cyst-like papule	NP	Solid BCC (1)	Surgery
Leibovitch I et al. (2005) [52]	Orbital invasion by periocular basal cell carcinoma	Case series	Ophthalmology	64	M (49) F (15)	Range 35–93 (70/72)	NA	Upper eyelid Lower eyelid Medial canthus Lateral canthus	Eyelid Periocular area	Nodule	NP	Infiltrative (33) Morpheaform/sclerosing (20) Nodular (2) Micronodular (2) Superficial (2)	Surgery ± RT RT alone
Ben Simon G.J et al. (2005) [53]	Orbital exenteration: one size does not fit all	Case series	Ophthalmology	6 (among 34 cases)	NA	Range 37–93 (67/NA)	NA	Orbit	Ocular area	NA	NP	BCC (6)	Surgery
Cameron M et al. (2005) [54]	Synchronous reconstruction of the exenterated orbit with a pericranial flap, skin graft and osseointegrated implants	Case reports	Maxillofacial surgery	1	M (1)	74	NA	Medial canthus	Periocular area	NA	NP	Morpheaform BCC (1)	Surgery
Moretti E et al. (2005) [55]	Complete mobilization of the cheek zone for orbit exenteration	Case series	Plastic surgery	7 (among 10 cases)	M (5) F (2)	Range 62–85 (73.1/73)	NA	Eyelid Canthus Cornea	Eyelid Ocular area Periocular area	NA	NP	BCC (7)	Surgery
Khandwala MA et al. (2005) [56]	Outcome of periocular basal cell carcinoma managed by overnight paraffin section	Retrospective study	Ophthalmology	93	NA	NA	NA	Upper eyelid Lower eyelid Medial canthus Lateral canthus	Eyelid Periocular area	NA	NP	Nodular BCC (82) Morpheaform BCC (8) Multifocal BCC (3)	Surgery
Papadopoulos O et al. (2005) [57]	Orbitopalpebral repair after 835 excisions of malignant tumours	Retrospective study	Plastic surgery	714 (among 835 cases)	NA	NA	NA	Upper eyelid Lower eyelid Medial canthus Lateral canthus	Eyelid Periocular area	NA	NP	BCC (714)	Surgery
Leibovitch I et al. (2005) [58]	Incidence of host site complications in periocular full thickness skin grafts	Prospective case series	Ophthalmology	368 (among 397 cases)	NA	Range 20–91 (60/NA)	NA	Upper eyelid Lower eyelid Medial canthus	Eyelid Periocular area	NA	NP	BCC (368)	Surgery
Mencía-Gutiérrez E et al. (2005) [59]	Lacrimal caruncle primary basal cell carcinoma: case report and review	Case reports + review	Ophthalmology	1	M (1)	80	Hypertension and heart attack (1)	Caruncle	Caruncle	Pigmented lesion with fine, superficial vascularization	NP	Nodular BCC (1)	Surgery
Mavrikakis I et al. (2006) [60]	Linear basal cell carcinoma: a distinct clinical entity in the periocular region	Case series	Oculoplastic surgery	4	M (2) F (2)	Range 68–79 (74/74.5)	NA	Lower eyelid	Eyelid	NA	NP	Pigmented nodular/infiltrative BCC (2) Nodular BCC (2)	Surgery
Rossman D et al. (2006) [61]	Basal cell carcinoma of the caruncle	Case reports	Ophthalmology	1	M (1)	84	Skin cancer (1)	Caruncle	Caruncle	Whitish-yellow nodule	NP	Nodular BCC (1)	Surgery + RT
Kaeser PF et al. (2006) [62]	Tumors of the caruncle: a clinicopathologic correlation	Case series	Ophthalmology	2 (among 191 cases)	M (1) F (1)	Range 53–72 (62.5/62.5)	NA	Caruncle	Caruncle	NA	NP	BCC (1) Pigmented, macronodular BCC (1)	Surgery
Nemet AY et al. (2006) [63]	Management of periocular basal and squamous cell carcinoma: a series of 485 cases	Case series	Ophthalmology	417 (among 485 cases)	NA	Range 14–95 (68.3/NA)	BCC (NA) and SCC (NA)	Upper eyelid Lower eyelid Medial canthus Lateral canthus	Eyelid Periocular area	NA	NP	Nodular BCC (378) Morpheaform BCC (39)	Surgery ± RT
Taherian K et al. (2007) [64]	Surgical excision of periocular basal cell carcinomas	Case series	Ophthalmology	25	M (15) F (10)	Range 42–94 (76/NA)	NA	Upper eyelid Lower eyelid Medial canthus Lateral canthus	Eyelid Periocular area	Nodule	NP	BCC (25)	Surgery
Gericke A and Pitz S (2008) [65]	Maggot therapy for periocular skin graft failure in the immunocompromised patient	Letter	Ophthalmology	1 (among 2 cases)	M (1)	62	Heart transplant (1)	Medial canthus	Periocular area	NA	NP	BCC (1)	Surgery + Maggots therapy
Croce A et al. (2008) [66]	Orbital exenteration in elderly patients: personal experience	Case series	Otorhinolaryngology	4 (among 8 cases)	M (3) F (1)	Range 70–85 (77/76.5)	BCC (NA)	Eyelid Medial canthus Lateral canthus	Eyelid Periocular area	NA	NP	BCC (4)	Surgery
Kovacevic PT et al. (2009) [67]	Extended orbital exenteration in the treatment of advanced periocular skin cancer with primary reconstruction with a galeacutaneous flap	Retrospective study	Plastic surgery	14 (among 21 cases)	NA	Range 57–83 (68/NA)	NA	Eyelid Medial canthus Lateral canthus	Eyelid Periocular area	Ulcerated lesion	NP	BCC (14)	Surgery
Lawrence CM et al. (2009) [68]	Formalin-fixed tissue Mohs surgery (slow Mohs) for basal cell carcinoma: 5-year follow-up data	Prospective study	Dermatology	248 (among 1.336 cases)	NA	NA	NA	Upper eyelid Lower eyelid Medial canthus Lateral canthus	Eyelid Periocular area	NA	NP	BCC (248)	Surgery
Gayre GS et al. (2009) [69]	Outcomes of excision of 1750 eyelid and periocular skin basal cell and squamous cell carcinomas by modified en face frozen section margin-controlled technique	Case series	Ophthalmology	1.638 (among 12.862 cases)	NA	Range 15–96 (65/NA)	NA	Upper eyelid Lower eyelid Medial canthus Lateral canthus	Eyelid Periocular area	NA	NP	BCC (1.638)	Surgery
Lavaju P et al. (2009) [70]	Pattern of ocular tumors in the eastern region of Nepal	Retrospective study	Ophthalmology	15 (among 115 cases)	NA	NA	NA	Eyelid	Eyelid	NA	NP	BCC (15)	Surgery
Haefliger IO et al. (2009) [71]	Large upper eyelid full-thickness defects reconstructed only with an anterior lamella	Case reports	Ophthalmology	1 (among 3 cases)	M (1)	51	NA	Upper eyelid	Eyelid	NA	NP	BCC (1)	Surgery
Morris DS et al. (2009) [72]	Periocular basal cell carcinoma: 5-year outcome following Slow Mohs surgery with formalin-fixed paraffin-embedded sections and delayed closure	Prospective case series	Ophthalmology	278	M (137) F (141)	Range 25–89 (65.5/NA)	NA	Eyelid Periocular area	Eyelid Periocular area	NA	NP	Solid BCC (211) Morpheaform BCC (11) Superficial BCC (8) BCC (47)	Surgery
Levin F et al. (2009) [73]	Excision of periocular basal cell carcinoma with stereoscopic microdissection of surgical margins for frozen-section control: report of 200 cases	Case series	Ophthalmology	190 (among 200 cases)	NA	Range 31–95 (77/NA)	BCC (NA)	Upper eyelid Lower eyelid Medial canthus Lateral canthus	Eyelid Periocular area	Nodule Sclerosing lesion Superficial lesion	NP	BCC (190)	Surgery
Chadha V and Wright M (2009) [74]	Small margin excision of periocular basal cell carcinomas	Retrospective study	Ophthalmology	90	M (41) F (49)	Range 35–97 (73.9/NA)	NA	Upper eyelid Lower eyelid Medial canthus Lateral canthus	Eyelid Periocular area	NA	NP	Nodular BCC (87) Morpheaform BCC (3)	Surgery
Ong LY and Lane CM (2009) [75]	Eyelid contracture may indicate recurrent basal cell carcinoma, even after Mohs’ micrographic surgery	Case reports	Ophthalmology	3	M (2) F (1)	Range 53–92 (71.3/69)	Chronic obstructive airways disease and rectal carcinoma (1)	Upper eyelid Lower eyelid Medial canthus	Eyelid Periocular area	Nodule	NP	Infiltrative BCC (NA) Morpheaform BCC (NA)	Surgery + RT
Lee C et al. (2010) [76]	Primary ocular caruncolar basal cell carcinoma in a Chinese patient	Case reports	Ophthalmology	1	M (1)	73	Hypertension and coronary artery disease (1)	Caruncle	Caruncle	Pigmented nodule	NP	Pigmented BCC (1)	Surgery + MTX + Fluorouracil
Maheshwari R (2010) [77]	Review of orbital exenteration from an eye care centre in Western India	Retrospective study	Ophthalmology	3 (among 15 cases)	M (3)	Range 62–75 (66.7/63)	NA	Eyelid	Eyelid	NA	NP	BCC (3)	Surgery
Carneiro RC et al. (2010) [78]	Imiquimod 5% cream for the treatment of periocular basal cell carcinoma	Retrospective study	Ophthalmology	8	M (7) F (1)	Range 47–72 (63/65)	NA	Lower eyelid Medial canthus	Eyelid Periocular area	NA	NP	Nodular BCC (8)	Surgery + Imiquimod 5%
Garcia-Martin E et al. (2010) [79]	Efficacy and tolerability of imiquimod 5% cream to treat periocular basal cell carcinomas	Case series	Ophthalmology	15	M (10) F (5)	Range 53–84 (71.2/76)	Basal cell nevus syndrome (1); Aortic valve stenosis (1); Retinal detachment (1); Pituitary microadenoma (1); Atrial fibrillation (1)	Upper eyelid Lower eyelid Medial canthus Lateral canthus	Eyelid Periocular area	Growing mass	Superficial telangiectatic vessels Central umbilication	Nodular BCC (15)	Surgery + Imiquimod 5% + Cryotherapy or PDT
Madge SN et al. (2010) [80]	Globe-sparing surgery for medial canthal Basal cell carcinoma with anterior orbital invasion	Retrospective study	Ophthalmology	20	M (7) F (13)	Range 48–90 (72.6/78)	NA	Medial canthus	Periocular area	Palpable mass	NP	Infiltrative BCC (9) Nodular BCC (4) Micronodular BCC (2) Mixed BCC (4) Multifocal BCC (1)	Surgery + RT
Ross AH et al. (2010) [81]	The use of imiquimod in the treatment of periocular tumours	Case series	Ophthalmology	2 (among 5 cases)	F (2)	47, 75	NA	Lower eyelid	Eyelid	Erythematous plaque Nodule	NP	BCC (2)	Imiquimod 5%
Cannon PS et al. (2011) [82]	The ophthalmic side-effects of imiquimod therapy in the management of periocular skin lesions	Retrospective study	Ophthalmology	3 (among 47 cases)	NA	Range 42–95 (74/NA)	NA	Upper eyelid Lower eyelid Medial canthus Lateral canthus	Eyelid Periocular area	NA	NP	BCC (3)	Surgery + Imiquimod 5%
Alves L.F et al. (2011) [83]	Incidence of epithelial lesions of the conjunctiva in a review of 12,102 specimens in Canada	Retrospective study	Pathology	1 (among 12.102 cases)	NA	Range NA (59.9/NA)	NA	Conjunctiva	Ocular area	NA	NP	BCC (1)	NA
Shinder R et al. (2011) [84]	Survey of orbital tumors at a comprehensive cancer center in the United States	Retrospective study	Ophthalmology	7 (among 268 cases)	M (4) F (3)	Range 35–80 (NA/59)	NA	Orbit	Ocular area	NA	NP	BCC (7)	Surgery
Alfawaz AM and Al-Hussain HM (2011) [85]	Ocular manifestations of xeroderma pigmentosum at a tertiary eye care center in Saudi Arabia	Retrospective study	Ophthalmology	4 (among 27 cases)	NA	Range 5–67 (NA/19)	Xeroderma pigmentosum (4)	Eyelid	Eyelid	NA	NP	BCC (4)	Surgery
Gautam P et al. (2011) [86]	A profile of eye-lid conditions requiring reconstruction among the patients attending an oculoplasty clinic in mid-western region of Nepal	Retrospective study	Ophthalmology	5 (among 43 cases)	NA	Range NA (52/NA)	NA	Eyelid	Eyelid	NA	NP	BCC (5)	Surgery
Moesen I et al. (2011) [87]	Nitrous oxide cryotherapy for primary periocular basal cell carcinoma: outcome at 5 years follow-up	Prospective case series	Ophthalmology	95	M (37) F (58)	Range 44–92 (NA/71)	NA	Upper eyelid Lower eyelid Medial canthus Lateral canthus	Eyelid Periocular area	NA	NP	BCC (95)	Surgery + Nitrous oxide cryotherapy
Ichinokawa Y et al. (2011) [88]	Linear basal cell carcinoma: a case report	Case reports	Dermatology	1	F (1)	79	NA	Lower eyelid	Eyelid	Pigmented nodule	Arborizing vessels, multiple blue-grey globules, and large, blue-grey ovoid nests	Nodular and infiltrative BCC (1)	Surgery
Garcia-Martin E et al. (2011) [89]	Comparison of imiquimod 5% cream versus radiotherapy as treatment for eyelid basal cell carcinoma	Clinical trial	Ophthalmology	27	M (16) F (11)	Range 53–84 (73.8/NA)	NA	Eyelid	Eyelid	Nodule with telangiectatic vessels	NP	Nodular BCC (27)	Imiquimod 5% or RT
Gaitanis G et al. (2011) [90]	Imiquimod can be combined with cryosurgery (immunocryosurgery) for locally advanced periocular basal cell carcinomas	Letter	Dermatology	3	M (2) F (1)	Range 68–75 (70.7/69)	Hypertension, hyperlipidemia, chronic hepatitis B infection (1); Skin cancers (1)	Lower eyelid Medial canthus	Eyelid Periocular area	NA	NP	BCC (3)	Immunocryosurgery
Kadyan A et al. (2011) [91]	High rate of incomplete resection after primary excision of eyelid BCC: multi-staged resection rarely needs more than two procedures	Case series	Ophthalmology	90	M (36) F (54)	Range 47–98 (77/NA)	No comorbidities	Upper eyelid Lower eyelid Medial canthus Lateral canthus	Eyelid Periocular area	NA	NP	BCC (90)	Surgery
Kvannli L et al. (2012) [92]	The method of en face frozen section in clearing periocular basal cell carcinoma and squamous cell carcinoma	Retrospective study	Ophthalmology	204 (among 262 cases)	M (100) F (104)	Range 21–94 (65/NA)	NA	Upper eyelid Lower eyelid Medial canthus Lateral canthus	Eyelid Periocular area	NA	NP	Nodulocystic BCC (NA) Infiltrative BCC (NA) Morpheaform BCC (NA) Multifocal BCC (NA) Mixed pattern (NA)	Surgery
Ben Simon GJ et al. (2012) [93]	Histological and clinical features of primary and recurrent periocular basal cell carcinoma	Case series	Ophthalmology	87	M (52) F (35)	Range 33–96 (70/NA)	NA	Upper eyelid Lower eyelid Medial canthus Lateral canthus	Eyelid Periocular area	NA	NP	Solid, nodular or nodular ulcerative BCC (NA) Morpheaform or infiltrative BCC (NA) Cystic or nodulocystic BCC (NA)	Surgery
Bertelmann E and Rieck P (2012) [94]	Relapses after surgical treatment of ocular adnexal basal cell carcinomas: 5-year follow-up at the same university centre	Prospective study	Ophthalmology	366	M (167) F (199)	Range NA (69/NA)	Parkinson disease (1)	Upper eyelid Lower eyelid Medial canthus Lateral canthus	Eyelid Periocular area	NA	NP	Nodular BCC (NA) Ulcerative BCC (NA) Morpheaform BCC (NA)	Surgery
Hata M et al. (2012) [95]	Radiation therapy for primary carcinoma of the eyelid: tumor control and visual function	Retrospective study	Radiology	1 (among 23 cases)	NA	Range 60–91 (NA/79)	Gastric cancer (1)	Eyelid	Eyelid	NA	NP	BCC (1)	Surgery + RT
Meena M (2012) [96]	Triple-Flaps for lateral canthus reconstruction: a novel technique	Case reports	Ophthalmology	1	F (1)	45	NA	Eyelid Lateral canthus	Eyelid Periocular area	Pigmented nodule	NP	Nodular–ulcerative BCC (1)	Surgery
Attili SK et al. (2012) [97]	Role of non-surgical therapies in the management of periocular basal cell carcinoma and squamous intra-epidermal carcinoma: a case series and review of the literature	Case series + review	Dermatology	13 (among 22 cases)	M (9) F (4)	Range 59–79 (68.2/69)	Renal transplant (1)	Lower eyelid Medial canthus Lateral canthus	Eyelid Periocular area	NA	NP	Morpheic BCC (3) Nodulocystic BCC (7) Superficial BCC (3)	PDT or Imiquimod 5%
Iuliano A et al. (2012) [98]	Risk factors for orbital exenteration in periocular basal cell carcinoma	Case series	Ophthalmology	502 (among 506 cases)	NA	Range 47–84 (67.2/NA)	NA	Upper eyelid Lower eyelid Medial canthus Lateral canthus	Eyelid Periocular area	Nodule	NP	Infiltrative BCC (NA) Morpheaform BCC (NA) Nodular BCC (NA)	Surgery ± RT
Zeitouni NC et al. (2012) [99]	Orbital invasion by periocular infiltrating basal cell carcinoma	Case reports	Dermatology	1	M (1)	82	Hyperlipidemia, coronary artery disease, myocardial infarction (1)	Eyelid	Eyelid	NA	NP	BCC (1)	Surgery
Sirianni D et al. (2013) [100]	A 12-year retrospective survey of management of patients with malignant neoplasms in the orbital cavity in a Brazilian Cancer Hospital	Case series	Dentistry	156 (among 269 cases)	NA	NA	NA	Ocular	Ocular area	NA	NP	BCC (156)	Surgery
Fino P et al. (2013) [101]	First reported case of primary basal cell carcinoma of the right caruncle: a case report and review of the literature	Case reports + review	Plastic surgery	1	F (1)	24	No comorbidities	Caruncle	Caruncle	Pigmented lesion	NP	Solid BCC (1)	Surgery
Krema H et al. (2013) [102]	Orthovoltage radiotherapy in the management of medial canthal basal cell carcinoma	Retrospective study	Ophthalmology	90	M (39) F (51)	Range 48–97 (NA/73)	NA	Medial canthus	Periocular area	NA	NP	BCC (90)	RT or Surgery + RT
Bonavolontà G et al. (2013) [103]	An analysis of 2480 space-occupying lesions of the orbit from 1976 to 2011	Case series	Ophthalmology	87 (among 2.480)	NA	Range 42–75 (71/NA)	NA	Orbit	Ocular area	NA	NP	BCC (87)	Surgery
Yin VT et al. (2013) [104]	Targeted therapy for orbital and periocular basal cell carcinoma and squamous cell carcinoma	Case reports + review	Ophthalmology	1	M (1)	30	Basal cell nevus syndrome (1)	Upper eyelid Lower eyelid Medial canthus	Eyelid Periocular area	Nodule with telangiectasia	NP	BCC (1)	Surgery + Vismodegib
Sun MT et al. (2013) [105]	Periocular basal cell carcinoma pathological reporting	Letter	Ophthalmology	1618	NA	NA	NA	Upper eyelid Lower eyelid Medial canthus Lateral canthus	Eyelid Periocular area	NA	NP	BCC (1.618)	Surgery
Tullett M et al. (2013) [106]	Excision of periocular basal cell carcinoma guided by en face frozen section	Case series	Oculoplastic surgery	72 (among 78 cases)	NA	Range 35–97 (74.5/NA)	NA	Upper eyelid Lower eyelid Medial canthus	Eyelid Periocular area	NA	NP	Nodular BCC (NA) Micronodular BCC (NA) Superficial BCC (NA) Infiltrative BCC (NA)	Surgery
Gill HS et al. (2013) [107]	Vismodegib for periocular and orbital basal cell carcinoma	Prospective case series	Ophthalmology	7	M (5) F (2)	Range 43–101 (71/75)	SCC (2)	Upper eyelid Lower eyelid Lateral canthus	Eyelid Periocular area	Ulcerated lesion	NP	BCC (7)	Surgery + Vismodegib
Litwin AS et al. (2013) [108]	Management of periocular basal cell carcinoma by Mohs micrographic surgery	Retrospective study	Ophthalmology	104	M (57) F (47)	Range 35–98 (66/NA)	NA	Upper eyelid Lower eyelid Medial canthus Lateral canthus	Eyelid Periocular area	NA	NP	Nodular BCC (67) Nodular/infiltrative or nodular/superficial BCC (13) Infiltrative BCC (15) Superficial BCC (9)	Surgery ± 5-fluorouracil or cryotherapy
Ho SF et al. (2013) [109]	5 Years review of periocular basal cell carcinoma and proposed follow-up protocol	Retrospective study	Ophthalmology	311 (among 412 cases)	NA	Range 28–99 (73.7/77)	No comorbidities	Upper eyelid Lower eyelid Medial canthus	Eyelid Periocular area	NA	NP	BCC (311)	Surgery
Pelosini L et al. (2013) [110]	In vivo optical coherence tomography (OCT) in periocular basal cell carcinoma: correlations between in vivo OCT images and postoperative histology	Prospective study	Ophthalmology	15 (among 16 cases)	NA	Range 49–91 (74/NA)	NA	Upper eyelid Lower eyelid Medial canthus Lateral canthus	Eyelid Periocular area	NA	NP	Nodular BCC (15)	Surgery
Avanaki MR et al. (2013) [111]	Investigation of basal cell carcinoma using dynamic focus optical coherence tomography	Case reports	Electrical engineering	3	M (3)	Range 75–78 (76.7/77)	NA	Eyelid	Eyelid	NA	NP	BCC (3)	Surgery
Woolley SD and Hughes C (2013) [112]	A young military pilot presents with a periocular basal cell carcinoma: a case report	Case reports	Ophthalmology	1	M (1)	32	No comorbidities	Lower eyelid	Eyelid	Papule with telangiectasia	NP	Nodular BCC (1)	Surgery
Ugurlu S et al. (2014) [113]	Primary basal cell carcinoma of the caruncle: case report and review of the literature	Case reports + review	Ophthalmology	1	M (1)	67	No comorbidities	Caruncle	Caruncle	Nodule	NP	Nodular BCC (1)	Surgery
Sardesai VR et al. (2014) [114]	Ocular myiasis with basal cell carcinoma	Case reports	Dermatology	1	M (1)	83	Hypertension, diabetes, ophthalmomyiasis (1)	Lateral canthus	Periocular area	Ulcerated lesion	NP	BCC (1)	Surgery
Salwa SP et al. (2014) [115]	Electrochemotherapy for the treatment of ocular basal cell carcinoma; a novel adjunct in the disease management	Case reports	Oculoplastic surgery	3	F (3)	Range 84–98 (91/90)	Multiple (NA)	Eyelid	Eyelid	NA	NP	BCC (3)	Electrochemotherapy with Bleomycin
Halkud R et al. (2014) [116]	Xeroderma pigmentosum: clinicopathological review of the multiple oculocutaneous malignancies and complications	Case series	Surgery	8 (among 11 cases)	M (3) F (5)	NA	Xeroderma pigmentosum (8)	Eyelid	Eyelid	NA	NP	BCC (8)	Surgery
Das D et al. (2014) [117]	Profile of ocular and adnexal tumours at a Tertiary Institute of Northeast India	Retrospective study	Pathology	33 (among 1.003 cases)	NA	NA	NA	Eyelid	Eyelid	NA	NP	BCC (33)	Surgery
Qassemyar A et al. (2014) [118]	Orbital exenteration and periorbital skin cancers	Retrospective study	Plastic surgery	8 (among 26 cases)	M (5) F (3)	Range 19–89 (68/NA)	NA	Upper eyelid Lower eyelid Lateral canthus	Eyelid Periocular area	NA	NP	BCC (8)	Surgery
Berenji F et al. (2014) [119]	A case of secondary ophthalmomyiasis caused by chrysomya bezziana (Diptera: Calliphorid)	Case reports	Parasitology	1	F (1)	55	Ophthalmomyiasis (1)	Upper eyelid	Eyelid	Eroded lesion	NP	BCC (1)	Surgery
Wu A et al. (2014) [120]	Histological subtypes of periocular basal cell carcinoma	Retrospective study	Ophthalmology	1713	M (966) F (747)	Range 21–101 (68.8/71)	NA	Upper eyelid Lower eyelid Medial canthus Lateral canthus	Eyelid Periocular area	NA	NP	Nodular BCC (NA) Infiltrative BCC (NA) Superficial BCC (NA) Micronodular BCC (NA)	Surgery
Goyal S et al. (2014) [121]	Nonhealing trauma masking periocular basal cell carcinoma in a young black male	Letter	Ophthalmology	1	M (1)	39	No comorbidities	Medial canthus	Periocular area	Ulcerated papule	NP	BCC (1)	Surgery
Ebrahimi A et al. (2014) [122]	Superpulsed CO_2_ laser with intraoperative pathologic assessment for treatment of periorbital basal cell carcinoma involving eyelash line	Case series	Dermatology	20	M (13) F (7)	Range 42–80 (61.4/NA)	NA	Upper eyelid Lower eyelid Medial canthus Lateral canthus	Eyelid Periocular area	Nodule Pigmented lesion Superficial lesion	NP	Solid BCC (15) Cystic BCC (2) Infiltrative BCC (1) Micronodular BCC (1)	CO_2_ laser
Cinotti E et al. (2014) [123]	The role of in vivo confocal microscopy in the diagnosis of eyelid margin tumors: 47 cases	Case series	Dermatology	14 (among 45 cases)	M (7) F (7)	Range 13–86 (70/75.5)	NA	Eyelid	Eyelid	Nodule Papule Macule	NP	BCC (14)	Surgery
De Macedo EM et al. (2015) [124]	Imiquimod cream efficacy in the treatment of periocular nodular basal cell carcinoma: a non-randomized trial	Prospective case series	Ophthalmology	19	M (13) F (6)	Range 60–>70 (NA/NA)	NA	Lower eyelid Medial canthus	Eyelid Periocular area	Erythematous lesion	NP	BCC (19)	Surgery + Imiquimod 5%
Eftekhari K et al. (2015) [125]	Local recurrence and ocular adnexal complications following electronic surface brachytherapy for basal cell carcinoma of the lower eyelid	Case reports	Oculoplastic surgery	1	M (1)	60	NA	Lower eyelid	Eyelid	Pearly lesion	NP	BCC (1)	Surgery
Sen S et al. (2015) [126]	Impression cytology diagnosis of ulcerative eyelid malignancy	Prospective case series	Pathology	13 (among 32 cases)	NA	Range 22–87 (60.7/NA)	NA	Eyelid	Eyelid	NA	NP	BCC (13)	Surgery
Suarez MJ et al. (2015) [127]	Clinicopathological features of ophthalmic neoplasms arising in the setting of xeroderma pigmentosum	Case series	Pathology	1 (among 6 cases)	M (1)	25	Xeroderma pigmentosum (1)	Lower eyelid	Eyelid	Nodule	NP	Nodular BCC (1)	Surgery
Meyer D and Gooding C (2015) [128]	Intralesional bleomycin as an adjunct therapeutic modality in eyelid and extraocular malignancies and tumors	Case series	Ophthalmology	3 (among 4 cases)	M (2) F (1)	Range 80–92 (84.3/81)	Angina (1)	Lower eyelid	Eyelid	NA	NP	BCC (3)	Intralesional bleomycin
Chen JJ et al. (2015) [129]	Review of ocular manifestations of nevoid basal cell carcinoma syndrome: what an ophthalmologist needs to know	Case reports + review	Ophthalmology	1	M (1)	31	Basal cell nevus syndrome (1)	Lower eyelid	Eyelid	Ulcerated lesions	NP	Nodular BCC (1)	Surgery
Domingo RE et al. (2015) [130]	Tumors of the eye and ocular adnexa at the Philippine Eye Research Institute: a 10-year review	Retrospective study	Ophthalmology	60 (among 1.551 cases)	M (25) F (35)	NA	NA	Eyelid Conjunctiva Orbit	Ocular area Periocular area	NA	NP	BCC (60)	Surgery
Demirci H et al. (2015) [131]	Efficacy of Vismodegib (Erivedge^TM^) for basal cell carcinoma involving the orbit and periocular area	Case series	Ophthalmology	8	M (6) F (2)	Range 60–86 (71/69)	Basal cell nevus syndrome (1); BCC (NA)	Upper eyelid Lower eyelid Medial canthus Lateral canthus Orbit	Eyelid Ocular area Periocular area	NA	NP	BCC (8)	Vismodegib ± Surgery
Ozgur OK et al. (2015) [132]	Hedgehog pathway inhibition for locally advanced periocular basal cell carcinoma and basal cell nevus syndrome	Case series	Ophthalmology	8 (among 12 cases)	M (7) F (1)	Range 33–86 (65/66.5)	Basal cell nevus syndrome (1)	Upper eyelid Lower eyelid Medial canthus Lateral canthus Orbit	Eyelid Ocular area Periocular area	NA	NP	BCC (8)	Vismodegib ± Another Hedgehog pathway inhibitor
Sun MT et al. (2015) [133]	Accuracy of biopsy in subtyping periocular basal cell carcinoma	Retrospective study	Ophthalmology	167	NA	NA	NA	Upper eyelid Lower eyelid Medial canthus Lateral canthus	Eyelid Periocular area	NA	NP	Nodular BCC (NA) Superficial BCC (NA) Mixed BCC (NA)	Biopsy
Tan E et al. (2015) [134]	Growth of periocular basal cell carcinomas	Observational study	Dermatology	113 (among 115 cases)	NA	Range 61–79 (69/NA)	Skin cancer; immunosuppression (NA)	Upper eyelid Lower eyelid Medial canthus Lateral canthus	Eyelid Periocular area	NA	NP	Nodular BCC (30) Infiltrative BCC (19) Superficial BCC (19) Micronodular BCC (19) Other types of BCC	Surgery
Pelosini L et al. (2015) [135]	A novel imaging approach to periocular basal cell carcinoma: in vivo optical coherence tomography and histological correlates	Prospective study	Ophthalmology	15	M (12) F (3)	Range 49–91 (74/79)	NA	Upper eyelid Lower eyelid Medial canthus Lateral canthus	Eyelid Periocular area	NA	NP	Nodular BCC (15)	Surgery
Nikose A et al. (2016) [136]	Periocular basal cell carcinoma in a young school teacher	Case reports	Ophthalmology	1	F (1)	34	No comorbidities	Lateral canthus	Periocular area	Pigmented nodule	NP	Adenoid BCC (1)	Surgery
Hamroush A and Cheung D (2016) [137]	Irregularly luscious lashes: difficult to say but a sinister sign to miss	Case reports	Ophthalmology	1	M (1)	84	Glaucoma, pseudophakia, dry age-related macular degeneration (AMD), macular atrophy, hypertension and hypercholesterolemia (1)	Lower eyelid	Eyelid	NA	NP	Micronodular BCC (1)	Surgery
Szewczyk M et al. (2016) [138]	Basal cell carcinoma in farmers: an occupation group at high risk	Retrospective study	Surgery	36 (among 312 cases)	M (22) F (14)	Range 25–>80 (NA/NA)	NA	Eyelid	Eyelid	NA	NP	BCC (36)	Surgery
Pandey TR et al. (2016) [139]	A case of orbital myiasis in recurrent eyelid basal cell carcinoma invasive into the orbit	Case reports	Ophthalmology	1	F (1)	73	Myiasis (1)	Orbit	Ocular area	Ulcerated nodule	NP	BCC (1)	Surgery
Bălăşoiu AT et al. (2016) [140]	Assessment of VEGF and EGFR in the study of angiogenesis of eyelid carcinomas	Retrospective study	Pathology	23 (among 43 cases)	NA	NA	NA	Eyelid	Eyelid	NA	NP	BCC (23)	Surgery
Treacy MP et al. (2016) [141]	Mohs micrographic surgery for periocular skin tumours in Ireland	Retrospective study	Ophthalmology	107 (among 127 cases)	NA	Range 28–91 (68/NA)	NA	Upper eyelid Lower eyelid Medial canthus	Eyelid Periocular area	NA	NP	Nodular BCC (68) Infiltrative BCC (26) Nodulocystic BCC (8) Adenoid BCC (5) Multifocal BCC (3) Superficial BCC (2)	Surgery
Wilson ME et al. (2016) [142]	Acute Charles Bonnet syndrome following Hughes procedure	Case reports	Ophthalmology	1	M (1)	69	Hypertension (1)	Lower eyelid	Eyelid	Erythematous lesion	NP	Morpheaform BCC (1)	Surgery
Sin CW et al. (2016) [143]	Recurrence rates of periocular basal cell carcinoma following Mohs micrographic surgery: a retrospective study	Case series	Oculoplastic surgery	390	M (170) F (220)	Range 24–96 (67/68)	NA	Upper eyelid Lower eyelid Medial canthus Lateral canthus	Eyelid Periocular area	NA	NP	Nodular BCC (178) Morpheaform BCC (105) Infiltrative BCC (16) Micronodular BCC (5) Pigmented BCC (4) Superficial BCC (4) Combined subtypes (2) BCC (76)	Surgery + Curettage or cryotherapy or RT
Gaitanis G et al. (2016) [144]	Cryosurgery during imiquimod (Immunocryosurgery) for periocular basal cell carcinomas: an efficacious minimally invasive treatment alternative	Retrospective study	Dermatology	14 (among 16 cases)	M (6) F (8)	Range 52–85 (74.9/79)	NA	Lower eyelid Medial canthus Lateral canthus	Eyelid Periocular area	NA	NP	Nodular BCC (14)	Immunocryosurgery
Celebi AR et al. (2016) [145]	Evaluation of the “Hedgehog” signalling pathways in squamous and basal cell carcinomas of the eyelids and conjunctiva	Retrospective study	Ophthalmology	41 (among 75 cases)	M (23) F (18)	Range 29–78 (60.7/NA)	NA	Eyelid	Eyelid	NA	NP	BCC (41)	Surgery
Lin SY et al. (2016) [146]	TERT promoter mutations in periocular carcinomas: implications of ultraviolet light in pathogenesis	Retrospective study	Pathology	20	NA	NA	NA	Eyelid Medial canthus	Eyelid Periocular area	NA	NP	Nodular BCC (17) Infiltrative BCC (1) Micronodular BCC (1) Morpheaform BCC (1)	Surgery
Yordanov YP and Shef A (2016) [147]	Synchronous basal cell carcinoma of the inferior eyelid-combined surgical approach for single-stage ablation	Case reports	Plastic surgery	1	F (1)	67	NA	Lower eyelid	Eyelid	Exophytic pigmented tumor Pigmented plaque	NP	Pigmented BCC (2)	Surgery
Zlatarova ZI et al. (2016) [148]	Eyelid reconstruction with full thickness skin grafts after carcinoma excision	Retrospective study	Ophthalmology	35 (among 39 cases)	NA	Range 26–95 (71/NA)	NA	Upper eyelid Lower eyelid Medial canthus	Eyelid Periocular area	NA	NP	BCC (35)	Surgery
Gerring RC et al. (2017) [149]	Orbital exenteration for advanced periorbital non-melanoma skin cancer: prognostic factors and survival	Case series	Otorhinolaryngology	20 (among 49 cases)	NA	Range 39–85 (70.3/NA)	Skin cancer (NA)	Upper eyelid Lower eyelid Medial canthus Lateral canthus	Eyelid Periocular area	NA	NP	BCC (20)	Surgery
Shafi F et al. (2017) [150]	Medial canthal defects following tumour excision: To reconstruct or not to reconstruct?	Retrospective study	Oculoplastic surgery	63 (among 68 cases)	NA	Range 38–94 (70.5/NA)	NA	Medial canthus	Periocular area	NA	NP	BCC (63)	Surgery
Agarwal R et al. (2017) [151]	Bilateral ocular surface squamous neoplasia with bilateral periocular basal cell carcinoma in a case of xeroderma pigmentosum	Case reports	Ophthalmology	1	M (1)	9	Xeroderma pigmentosum (1)	Medial canthus	Periocular area	Eroded papule-nodule	NP	BCC (1)	Surgery + Imiquimod 5%
Khan L. et al. (2017) [152]	Conjunctival lesions: when should we perform biopsy?	Retrospective study	Ophthalmology	1 (among 129 cases)	NA	Range 25–75 (NA/NA)	NA	Eyelid	Eyelid	NA	NP	BCC (1)	Surgery
Shields CL et al. (2017) [153]	Conjunctival tumors in 5002 cases. Comparative analysis of benign versus malignant counterparts	Case series	Ophthalmology	6 (among 5002 cases)	M (3) F (3)	Range 21–>60 (NA/NA)	NA	Eyelid Caruncle	Eyelid Caruncle	NA	NP	BCC (6)	Surgery
Mutaf M and Temel M (2017) [154]	A new technique for total reconstruction of the lower lid	Case series	Plastic surgery	22 (among 24 cases)	NA	Range 45–72 (56.8/NA)	NA	Eyelid Canthus	Eyelid Periocular area	NA	NP	BCC (22)	Surgery
Kiratli H and Koç I (2017) [155]	Orbital exenteration: Institutional review of evolving trends in indications and rehabilitation techniques	Retrospective study	Ophthalmology	10 (among 100 cases)	M (7) F (3)	Range NA (56.9/NA)	NA	Eyelid Conjunctiva	Eyelid Ocular area	NA	NP	BCC (10)	Surgery
Furdova A and Lukacko P (2017) [156]	Periocular basal cell carcinoma predictors for recurrence and infiltration of the orbit	Case series	Ophthalmology	7 (among 256 cases)	NA	Range 52–82 (58/NA)	NA	Lower eyelid Medial canthus	Eyelid Periocular area	NA	NP	BCC (7)	Surgery ± Brachytherapy
Wong KY et al. (2017) [157]	Vismodegib for locally advanced periocular and orbital basal cell carcinoma: a review of 15 consecutive cases	Case series	Plastic surgery	15	M (9) F (6)	Range 44–90 (74/78)	Basal cell nevus syndrome (1)	Upper eyelid Lower eyelid Medial canthus Lateral canthus	Eyelid Periocular area	NA	NP	BCC (15)	Surgery + Vismodegib + RT
Tan E et al. (2017) [158]	A practical decision-tree model to predict complexity of reconstructive surgery after periocular basal cell carcinoma excision	Prospective study	Dermatology	150 (among 162 cases)	NA	Range NA (NA/72)	Skin cancer; immunosuppression (NA)	Upper eyelid Lower eyelid Medial canthus Lateral canthus	Eyelid Periocular area	NA	NP	Nodular BCC (NA) Infiltrative BCC (NA) Superficial BCC (NA) Micronodular BCC (NA) Adenoid cystic BCC (NA) BCC (NA)	Surgery
Papastefanou VP and René C (2017) [159]	Secondary resistance to Vismodegib after initial successful treatment of extensive recurrent periocular basal cell carcinoma with orbital invasion	Case reports	Ophthalmology	1	M (1)	84	NA	Upper eyelid	Eyelid	Ulcerated lesion	NP	BCC (1)	Vismodegib + Surgery
Karabulut GO et al. (2017) [160]	Imiquimod 5% cream for the treatment of large nodular basal cell carcinoma at the medial canthal area	Case reports	Ophthalmology	3	NA	Range 45–73 (55.7/49)	NA	Medial canthus	Periocular area	NA	NP	Nodular BCC (3)	Surgery + Imiquimod 5%
Fatigato G et al. (2017) [161]	Risk factors associated with relapse of eyelid basal cell carcinoma: results from a retrospective study of 142 patients	Retrospective study	Plastic surgery	142 (among 205 cases)	NA	Range 34–96 (70/NA)	NA	Upper eyelid Lower eyelid Medial canthus Lateral canthus	Eyelid Periocular area	NA	NP	BCC (142)	Surgery
O’Halloran L et al. (2017) [162]	Periocular Mohs micrographic surgery in Western Australia 2009–2012: A single centre retrospective review and proposal for practice benchmarks	Retrospective study	Dermatology	589 (among 690 cases)	NA	Range 23–93 (65/NA)	NA	Upper eyelid Lower eyelid Medial canthus Lateral canthus	Eyelid Periocular area	NA	NP	BCC (589)	Surgery
Bladen JC et al. (2018) [163]	Analysis of hedgehog signalling in periocular sebaceous carcinoma	Observational study	Ophthalmology	15 (among 30 cases)	NA	Range NA (73/NA)	NA	Eyelid	Eyelid	NA	NP	Nodular BCC (15)	Surgery
Tchernev G et al. (2018) [164]	Locally advanced basal cell carcinoma with intraocular invasion	Case reports	Dermatology	1	M (1)	103	NA	Upper eyelid	Eyelid	Ulcerated papule	NP	BCC (1)	Surgery
Kaiser U (2018) [165]	Polarization and distribution of tumor-associated macrophages and COX-2 expression in basal cell carcinoma of the ocular adnexae	Retrospective study	Ophthalmology	30	M (11) F (19)	Range 49–97 (75/NA)	NA	Upper eyelid Lower eyelid Medial canthus Lateral canthus	Eyelid Periocular area	NA	NP	Nodular BCC (15) Fibrosing BCC (15)	Surgery
Khardenavis SJ et al. (2018) [166]	Ophthalmomyiasis in a case of basal cell carcinoma of eyelid	Case reports	Ophthalmology	1	F (1)	74	Ophthalmomyiasis (1)	Lower eyelid	Eyelid	Ulcerated lesion	NP	Infiltrating BCC (1)	Ampicillin–Sulbactam + Surgery + Ivermectin
Yunoki T et al. (2018) [167]	Gene networks in basal cell carcinoma of the eyelid, analysed using gene expression profiling	Case reports	Ophthalmology	2	F (2)	78, 83	NA	Eyelid	Eyelid	NA	NP	BCC (2)	Surgery
Al Wohaib M et al. (2018) [168]	Characteristics and factors related to eyelid basal cell carcinoma in Saudi Arabia	Retrospective study	Ophthalmology	129	M (76) F (53)	Range 16–105 (NA/71.0)	Xeroderma pigmentosum (1)	Upper eyelid Lower eyelid Medial canthus Lateral canthus	Eyelid Periocular area	Ulcerated lesion Pigmented lesion Nodular lesion Mixed	NP	Nodular BCC (65) Ulcerative BCC (17) Morpheaform BCC (14) Sclerosing BCC (5) Mixed BCC (4)	Surgery
Hogarty DT et al. (2018) [169]	Vismodegib and orbital excision for treating locally advanced basal cell carcinoma	Case reports	Ophthalmology	1	F (1)	51	NA	Medial canthus	Periocular area	NA	NP	Nodular BCC (1)	Surgery + RT + Vismodegib
Espi P et al. (2018) [170]	Clinical and genetic characteristics of xeroderma pigmentosum in Nepal	Retrospective study	Dermatology	1 (among 17 cases)	M (1)	24	Xeroderma pigmentosum (1)	Ocular	Ocular area	NA	NP	BCC (1)	Surgery
Damasceno JC et al. (2018) [171]	Largest case series of Latin American eyelid tumors over 13 years from a single center in Sao Paulo, Brazil	Retrospective study	Ophthalmology	226 (among 1.113 cases)	NA	Range NA (NA/65)	NA	Eyelid	Eyelid	NA	NP	BCC (226)	Surgery
Lemaître S et al. (2018) [172]	Outcomes after surgical resection of lower eyelid tumors and reconstruction using a nasal chondromucosal graft and an upper eyelid myocutaneous flap	Retrospective study	Ophthalmology	17 (among 25 cases)	NA	Range 21–91 (72/NA)	Xeroderma Pigmentosum (1); Conjunctival mucosa-associated lymphoid tissue lymphoma (1)	Eyelid	Eyelid	NA	NP	Superficial BCC (2) Nodular BCC (3) Morpheic and nodular BCC (1) Morpheic and infiltrating BCC (2) Morpheic BCC (1) Infiltrating BCC (4) Infiltrating and nodular BCC (4)	Surgery
Costan VV et al. (2018) [173]	Mixed (nodular and morpheic) upper eyelid basal cell carcinoma with orbital invasion–histological and clinical features	Case reports	Maxillofacial surgery	1	M (1)	63	Dermatomyositis Hypertension (1)	Upper eyelid	Eyelid	Ulcerated lesion	NP	Nodular/morpheic BCC (1)	Surgery
Martell K et al. (2018) [174]	Radiation therapy for deep periocular cancer treatments when protons are unavailable: is combining electrons and orthovoltage therapy beneficial?	Case series	Oncology	3 (among 4 cases)	NA	NA	NA	Orbit Medial canthus	Ocular area Periocular area	NA	NP	BCC (3)	Electrons and orthovoltage therapy
Li X et al. (2019) [175]	Ocular preservation through limited tumor excision combined with ALA-PDT in patients with periocular basal cell carcinoma	Case series	Plastic surgery	8	M (6) F (2)	Range 54–81 (NA/NA)	NA	Eyelid	Eyelid	NA	NP	BCC (8)	Surgery + PDT
Sagiv O et al. (2019) [176]	Ocular preservation with neoadjuvant Vismodegib in patients with locally advanced periocular basal cell carcinoma	Case series	Oculoplastic surgery	8	M (8)	Range 55–84 (69/69)	NA	Upper eyelid Lower eyelid Medial canthus Lateral canthus	Eyelid Periocular area	NA	NP	Nodular BCC (1) Nodular and infiltrative BCC pattern (7)	Vismodegib + Surgery
McGrath L.A et al. (2019) [177]	Staged excision of primary periocular basal cell carcinoma: absence of residual tumour in re- excised specimens: a 10-year series	Retrospective study	Ophthalmology	120	M (56) F (64)	Range 38–94 (75/77)	Cancers or autoimmune diseases (6); History of other skin cancers (34)	Upper eyelid Lower eyelid Medial canthus Lateral canthus	Eyelid Periocular area	NA	NP	Nodular BCC (29) Superficial BCC (12) Micronodular (66) Infiltrative/ Morpheic (13)	Surgery
Klyuchareva SV et al. (2019) [178]	Treatment of basal cell cancer with a pulsed copper vapor laser: a case series	Case series	Dermatology	8	M (2) F (6)	Range 34–77 (53.1/50.5)	NA	Upper eyelid Lower eyelid Medial canthus	Eyelid Periocular area	NA	NP	Nodular BCC (7) Infiltrative BCC (1)	Surgery + Dual-wavelength copper vapor laser
Kaliki S et al. (2019) [179]	Ocular and periocular tumors in xeroderma pigmentosum: a study of 120 Asian Indian patients	Case series	Ophthalmology	8 (among 120 cases)	NA	Range 1–53 (19/18)	Xeroderma pigmentosum (8)	Lower eyelid Lateral canthus	Eyelid Periocular area	NA	NP	BCC (8)	Surgery
Cohen S et al. (2019) [180]	The Amazon Ocular Oncology Center: the first three years	Retrospective study	Ophthalmology	4 (among 221 cases)	NA	NA	NA	Eyelid	Eyelid	NA	NP	BCC (4)	NA
Vavassori A et al. (2019) [181]	Mould-based surface high-dose-rate brachytherapy for eyelid carcinoma	Retrospective study	Radiotherapy	5 (among 9 cases)	M (3) F (2)	Range 61–88 (74.4/77)	NA	Lower eyelid Medial canthus	Eyelid Periocular area	NA	NP	BCC (5)	Contact high-dose-rate brachytherapy
Melzer C et al. (2019) [182]	Basal cell carcinomas developing independently from BAP1-tumor predisposition syndrome in a patient with bilateral uveal melanoma: diagnostic challenges to identify patients with BAP1-TPDS	Case reports	Ophthalmology	1	F (1)	69	BAP1-tumor predisposition syndrome (BAP1-TBDS), uveal melanoma, BCC, actinic keratosis and thyroid cancer (1)	Upper eyelid Medial canthus	Eyelid Periocular area	Telangiectatic lesion Ulcerated lesion	NP	BCC (69)	Surgery
Alam MS et al. (2019) [183]	Sensitivity and specificity of frozen section diagnosis in orbital and adnexal malignancies	Retrospective study	Oculoplastic surgery	6 (among 55 cases)	NA	Range 0.3–81 (51.5/NA)	NA	Eyelid	Eyelid	NA	NP	BCC (6)	Surgery
Bergeron S et al. (2019) [184]	Novel application of anterior segment optical coherence tomography for periocular imaging	Prospective study	Ophthalmology	38 (among 50 cases)	M (18) F (20)	Range 27–92 (70/75)	NA	Upper eyelid Lower eyelid Medial canthus Lateral canthus	Eyelid Periocular area	Telangiectatic nodule Telangiectatic plaque	Scar-like depigmentation and vessels with a crown distribution or irregularly branched	BCC (38)	Surgery
Mathis J et al. (2019) [185]	Oral hedgehog pathway inhibition as a means for ocular salvage in locally advanced intraorbital basal cell carcinoma	Case reports + review	Dermatology	1	M (1)	60	NA	Medial canthus	Periocular area	NA	NP	BCC (1)	Vismodegib
Costea CF et al. (2019) [186]	Periocular basal cell carcinoma: demographic, clinical, histological and immunohistochemical evaluation of a series of 39 cases	Retrospective study	Ophthalmology	36 (among 39 cases)	NA	Range 26–87 (66/NA)	NA	Upper eyelid Lower eyelid Medial canthus	Eyelid Periocular area	Ulcerated nodule	NP	BCC (36)	Surgery
Weesie F et al. (2019) [187]	Recurrence of periocular basal cell carcinoma and squamous cell carcinoma after Mohs micrographic surgery: a retrospective cohort study	Retrospective study	Dermatology	683 (among 729 cases)	M (293) F (390)	Range 58–77 (NA/69)	NA	Upper eyelid Lower eyelid Medial canthus Lateral canthus	Eyelid Periocular area	NA	NP	Superficial BCC (14) Nodular BCC (367) Micronodular BCC (43) Infiltrative BCC (256 BCC (3)	Surgery
González AR et al. (2019) [188]	Neoadjuvant Vismodegib and Mohs micrographic surgery for locally advanced periocular basal cell carcinoma	Retrospective study	Surgery	8	M (2) F (6)	Range 60–90 (76/74.5)	NA	Lower eyelid Medial canthus	Eyelid Periocular area	NA	NP	Nodular BCC (6) Infiltrative BCC (2)	Vismodegib ± Surgery
Khoo ABS et al. (2019) [189]	Comparative analyses of tumour volume doubling times for periocular and non-periocular head and neck basal cell carcinomas	Case series	Dermatology	47 (among 126 cases)	NA	NA	NA	Upper eyelid Lower eyelid Medial canthus Lateral canthus	Eyelid Periocular area	NA	NP	Nodular BCC (47)	Surgery
Kis EG et al. (2019) [190]	Electrochemotherapy in the treatment of locally advanced or recurrent eyelid-periocular basal cell carcinomas	Case series	Dermatology	12	M (7) F (5)	Range 11–86 (66.6/71)	Basal cell nevus syndrome (1); Xeroderma pigmentosum (1)	Upper eyelid Lower eyelid Medial canthus Lateral canthus	Eyelid Periocular area	NA	NP	BCC (12)	Surgery + Electrochemotherapy + Vismodegib
Sagiv O et al. (2019) [191]	Impact of Food and Drug Administration approval of Vismodegib on prevalence of orbital exenteration as a necessary surgical treatment for locally advanced periocular basal cell carcinoma	Retrospective study	Plastic surgery	42	M (31) F (11)	Range 43–90 (NA/66)	NA	Eyelid	Eyelid	NA	NP	BCC (42)	Surgery or RT or Vismodegib or palliative care
Eiger-Moscovich M et al. (2019) [192]	Efficacy of Vismodegib for the treatment of orbital and advanced periocular basal cell carcinoma	Retrospective study	Ophthalmology	21	M (16) F (5)	Range 59–91 (74/76)	Basal cell nevus syndrome (1); Melanoma (1); Paget disease (1)	Upper eyelid Lower eyelid Medial canthus Lateral canthus	Eyelid Periocular area	NA	NP	BCC (21)	Vismodegib + RT or Surgery + RT
Monheit G and Hrynewycz K (2019) [193]	Mohs surgery for periocular tumors	Case series	Dermatology	240 (among 289 cases)	NA	NA	NA	Upper eyelid Lower eyelid Medial canthus Lateral canthus	Eyelid Periocular area	NA	NP	Nodular BCC (182) Infiltrative BCC (58)	Surgery
Scofield-Kaplan SM et al. (2019) [194]	Predictive value of preoperative periocular skin cancer measurements for final Mohs defect size	Retrospective study	Ophthalmology	34 (among 42 cases)	NA	Range NA (68.5/NA)	NA	Lower eyelid Medial canthus Lateral canthus	Eyelid Periocular area	NA	NP	BCC (34)	Surgery
Brodowski R et al. (2019) [195]	Clinical-pathological characteristics of patients treated for cancers of the eyelid skin and periocular areas	Retrospective study	Maxillofacial surgery	213 (among 262 cases)	NA	Range <30–>81 (NA/NA)	NA	Upper eyelid Lower eyelid Medial canthus Lateral canthus	Eyelid Periocular area	Ulcerated nodule	NP	Nodular BCC (90) Ulcerative BCC (21) Cystic BCC (81) Cicatricial BCC (13) Pigmented (8)	Surgery
Mercuţ IM et al. (2020) [196]	The immunoexpression of MMP-1 and MMP-13 in eyelid basal cell carcinoma	Retrospective study	Ophthalmology	50	NA	NA	NA	Eyelid	Eyelid	NA	NP	Nodular BCC (41) Infiltrative BCC (9)	Surgery
Galindo-Ferreiro A et al. (2020) [197]	Characteristics and recurrence of primary eyelid basal cell carcinoma in Central Spain	Retrospective study	Ophthalmology	325 (among 337 cases)	NA	Range NA (69.4/NA)	NA	Upper eyelid Lower eyelid Medial canthus Lateral canthus	Eyelid Periocular area	NA	NP	Nodular BCC (215) Infiltrative BCC (48) Mixed BCC (39) Micronodular BCC (9) Superficial multifocal BCC (8) Sclerosing BCC (6)	Surgery
Hou X et al. (2020) [198]	Effective treatment of locally advanced periocular basal cell carcinoma with oral hedgehog pathway inhibitor?	Case reports	Ophthalmology	1	M (1)	73	BCC (1)	Lower eyelid	Eyelid	Morphoea-like lesion	NP	BCC (1)	Surgery + Sonidegib
Vijay V et al. (2020) [199]	Periocular basal cell carcinoma: 20-years experience at a tertiary Eye Care Center in South India	Case series	Ophthalmology	37 (among 185 cases)	M (19) F (18)	Range 37–90 (63.3/65)	Basal cell nevus syndrome (2)	Upper eyelid Lower eyelid Medial canthus Lateral canthus	Eyelid Periocular area	Pigmented and non-pigmented ulcerative lesion Pigmented and non-pigmented nodular mass Scar-like lesion	NP	Nodular BCC (1) Infiltrating BCC (1) Pigmented BCC (1)	Surgery
Ben Ishai M et al. (2020) [200]	Outcomes of Vismodegib for periocular locally advanced basal cell carcinoma from an open-label trial	Clinical trial	Ophthalmology	244	M (143) F (101)	Range NA (NA/72.0)	Multiple (NA)	Eyelid Canthus Lacrimal sac Orbit	Eyelid Ocular area Periocular area	NA	NP	BCC (244)	Vismodegib ± Surgery or RT
Fazil K et al. (2020) [201]	Evaluation of demographic features of eyelid lesions	Retrospective study	Ophthalmology	67 (among 86 cases)	NA	Range 7–88 (62.7/NA)	NA	Eyelid	Eyelid	NA	NP	BCC (67)	Surgery
Gupta R et al. (2020) [202]	Malignant tumors of the eyelid in India: a multicenter, multizone study on clinicopathologic features and outcomes	Retrospective study	Ophthalmology	47 (among 129 cases)	NA	Range 5–92 (62.7/NA)	NA	Eyelid	Eyelid	NA	NP	BCC (47)	Surgery
Boal NS et al. (2020) [203]	A black-pigmented eyelid nodule in an African American woman	Letter	Ophthalmology	1	F (1)	73	Hypertension, type 2 diabetes mellitus, hyperlipidemia (1)	Upper eyelid	Eyelid	Pigmented nodule	NP	Pigmented nodular BCC (1)	Surgery
Gąsiorowski K et al. (2020) [204]	Periocular basal cell carcinoma: recurrence risk factors/when to reoperate?	Retrospective study	Maxillofacial surgery	158	M (80) F (78)	Range 20–95 (68/NA)	NA	Upper eyelid Lower eyelid Medial canthus Lateral canthus	Eyelid Periocular area	NA	NP	BCC (158)	Surgery
Finskas O et al. (2020) [205]	Cryosurgery of periocular moderately aggressive basal cell carcinoma	Case series	Ophthalmology	53	M (14) F (39)	Range 49–97 (73/NA)	NA	Upper eyelid Lower eyelid Medial canthus Lateral canthus	Eyelid Periocular area	NA	NP	BCC (53)	Biopsy + Cryosurgery
Su MG et al. (2020) [206]	Treatment of periocular basal cell carcinoma with neoadjuvant Vismodegib	Case reports	Ophthalmology	1	F (1)	63	Cerebral aneurysm (1)	Medial canthus	Periocular area	Nodule	NP	BCC (1)	Vismodegib + Surgery
Oliphant H et al. (2020) [207]	Vismodegib for periocular basal cell carcinoma: an international multicentre case series	Case series	Ophthalmology	11 (among 13 cases)	M (6) F (5)	Range 43–91 (75.5/84)	NA	Upper eyelid Lower eyelid Medial canthus Lateral canthus	Eyelid Periocular area	Superficial lesion	NP	Nodular BCC (3) Infiltrative/nodular BCC (1) infiltrative BCC (3) Infiltrative-micronodular BCC (1) Superficial BCC (1) BCC (1) Cystic BCC (1)	Vismodegib ± Surgery ± RT
Herwig-Carl MC and Loeffler KU (2020) [208]	Regression of periocular basal cell carcinoma: a report of four cases with clinicopathologic correlation	Case series	Ophthalmology	4	M (2) F (2)	Range 74–87 (78.8/77)	Arterial hypertension (4); Cardiac arrhythmia (2); Hyperlipidemia (3); Diabetes mellitus type 2 Hypothyroidism (2); Gout (1); BCC (1)	Lower eyelid Medial canthus	Eyelid Periocular area	Telangiectatic ulcerated nodule	NP	BCC (4)	Surgery
MercuȚ IM et al. (2020) [209]	Histopathological features of the eyelid basal cell carcinomas	Retrospective study	Ophthalmology	92	NA	NA	NA	Eyelid	Eyelid	NA	NP	Nodular BCC (53) Infiltrative BCC (9) Superficial BCC (2) Micronodular BCC (8)	Surgery
Peden R et al. (2020) [210]	Small margin (up to 2 mm) excision of periocular basal cell carcinomas in the setting of a one-stop clinic-long-term outcomes at a minimum of 11 years’ follow-up	Case series	Ophthalmology	69	M (34) F (35)	Range 46–99 (74.9/77)	NA	Upper eyelid Lower eyelid Medial canthus Lateral canthus	Eyelid Periocular area	NA	NP	Morpheic BCC (8) Multifocal BCC (1) Nodulocystic BCC (1) Nodular BCC (2) BCC (57)	Surgery
Furdova A et al. (2020) [211]	Subtotal exenteration of the orbit for benign orbital disease	Case series	Ophthalmology	12 (among 14 cases)	NA	Range 67–80 (NA/73)	NA	Eyelid Medial canthus	Eyelid Periocular area	NA	NP	BCC (12)	Surgery
Ben Artsi E et al. (2020) [212]	Submental and anterior neck originated full-thickness skin grafts for periocular procedures	Case series	Oculoplastic surgery	3 (among 5 cases)	F (3)	Range 75–90 (83.3/85)	NA	Lower eyelid	Eyelid	NA	NP	BCC (3)	Surgery
Lin Z et al. (2021) [213]	A multicentre review of the histology of 1012 periocular basal cell carcinomas	Retrospective study	Ophthalmology	745 (among 1.012 cases)	NA	NA	NA	Upper eyelid Lower eyelid Medial canthus Lateral canthus	Eyelid Periocular area	NA	NP	Nodular BCC (NA) Infiltrative BCC (NA) Superficial BCC (NA)	Surgery
Stridh MT et al. (2021) [214]	Photoacoustic imaging of periorbital skin cancer ex vivo: unique spectral signatures of malignant melanoma, basal, and squamous cell carcinoma	Prospective study	Ophthalmology	8 (among 11 cases)	M (4) F (4)	Range 47–76 (65/69.5)	NA	Upper eyelid Lower eyelid Medial canthus Lateral canthus	Eyelid Periocular area	NA	NP	Nodular BCC (2) Infiltrative BCC (4) Morpheaform BCC (1) BCC (1)	Surgery
Adamski WZ et al. (2021) [215]	The prevalence of various eyelid skin lesions in a single-centre observation study	Retrospective study	Ophthalmology	110 (among 544 cases)	NA	Range 18–92 (60.5/NA)	NA	Upper eyelid Lower eyelid Medial canthus	Eyelid Periocular area	NA	NP	Nodular BCC (46) Ulcerative BCC (12) Superficial BCC (10) BCC (42)	Surgery
De Giorgi V et al. (2021) [216]	Treatment of periocular advanced basal cell carcinoma with Hedgehog pathway inhibitors: a single-center study and a new dedicated therapeutic protocol	Prospective study	Dermatology	15	M (7) F (8)	Range 63–94 (83/87)	NA	Lower eyelid Medial canthus Lateral canthus	Eyelid Periocular area	NA	NP	BCC (15)	Vismodegib or Sonidegib ± Surgery
Bergeron S et al. (2021) [217]	Optical coherence tomography of peri-ocular skin cancers: an optical biopsy	Prospective study	Ophthalmology	46 (among 58 cases)	NA	NA	NA	Periocular	Periocular area	NA	NP	BCC (46)	Surgery
Kaliki S and Das AV (2021) [218]	Ocular and periocular tumors in 855 Asian Indian geriatric patients	Retrospective study	Ophthalmology	25 (among 855 cases)	NA	Range 60–91 (68/67)	NA	Eyelid Orbit	Eyelid Periocular area	NA	NP	BCC (25)	Surgery
Shimizu N et al. (2021) [219]	Ten-year epidemiological study of ocular and orbital tumors in Chiba University Hospital	Retrospective study	Ophthalmology	15 (among 372 cases)	M (6) F (9)	Range NA (76.4/NA)	NA	Eyelid Conjunctiva	Eyelid Ocular area	NA	NP	BCC (15)	Surgery
Battista RA et al. (2021) [220]	Combination of Mustardè cheek advancement flap and paramedian forehead flap as a reconstructive option in orbital exenteration	Case reports	Otorhinolaryngology	1	NA	67	Chronic hepatopathy, severe obesity, kidney transplant under immunosuppressive therapy, arterial hypertension, cerebral ischemic events and conjunctival SCC (1)	Upper eyelid	Eyelid	NA	NP	BCC (1)	Surgery
Prídavková Z et al. (2021) [221]	Recurrent periocular basal cell carcinoma. Case Report	Case reports	Ophthalmology	1	M (1)	84	Cataract (1)	Upper eyelid	Eyelid	Ulcerated lesion	NP	BCC (1)	Surgery + EBRT
Rokohl AC and Heindl LM (2021) [222]	Effective systemic treatment of advanced periocular basal cell carcinoma with Sonidegib	Letter	Ophthalmology	1	M (1)	NA	Multiple BCC (1)	Lower eyelid	Eyelid	NA	NP	Morpheaform BCC (1)	Sonidegib
Almousa R (2021) [223]	Predictors for margin of resection >4 mm in the management of periocular basal cell carcinoma	Retrospective study	Ophthalmology	129 (among 142 cases)	NA	NA	NA	Upper eyelid Lower eyelid Medial canthus Lateral canthus	Eyelid Periocular area	NA	NP	BCC (129)	Surgery
Kahana A et al. (2021) [224]	Vismodegib for preservation of visual function in patients with advanced periocular basal cell carcinoma: the VISORB trial	Clinical trial	Ophthalmology	32 (among 34 cases)	NA	Range 48–95 (NA/68.5)	NA	Lower eyelid Medial canthus Lateral canthus	Eyelid Periocular area	NA	NP	BCC (32)	Vismodegib ± Surgery
Curragh DS et al. (2021) [225]	Neoadjuvant Vismodegib in the management of locally advanced periocular basal cell carcinoma	Case series	Ophthalmology	8	M (5) F (3)	Range 24–81 (57.6/57)	NA	Lower eyelid Medial canthus Lateral canthus	Eyelid Periocular area	NA	NP	Nodular BCC (4) Infiltrative BCC (3) Superficial/Nodular BCC (1)	Vismodegib + Surgery
Yazici B et al. (2021) [226]	Transnasal or transglabellar semicircular flap for medial canthal reconstruction	Case series	Ophthalmology	36 (among 38 cases)	NA	Range 41–87 (66/NA)	NA	Medial canthus	Periocular area	NA	NP	BCC (36)	Surgery
Weerdt G et al. (2021) [227]	Reconstruction of an extensive periocular and bilamellar defect of the lower and upper eyelid using local, regional and free chondral graft techniques: a case report	Case reports	Plastic surgery	1	F (1)	65	No comorbidities	Lower eyelid	Eyelid	NA	NP	BCC (1)	Surgery
Low KL et al. (2022) [228]	Primary basal cell carcinoma of the conjunctiva	Case reports	Ophthalmology	1	M (1)	67	No comorbidities	Conjunctiva	Ocular area	Pigmented nodule	NP	BCC (1)	Surgery + Topical mitomycin-C 0.02%
Ul Kadir SM et al. (2022) [229]	Clinicopathological analysis and surgical outcome of eyelid malignancies: a study of 332 cases	Case series	Ophthalmology	126 (among 332 cases)	NA	Range NA (64.6/NA)	NA	Upper eyelid Lower eyelid Medial canthus Lateral canthus	Eyelid Periocular area	NA	NP	BCC (126)	Surgery
Lemaıtre S et al. (2022) [230]	Total orbital exenteration with temporalis muscle transfer and secondary healing	Retrospective study	Ophthalmology	1 (among 29 cases)	NA	87	NA	Eyelid	Eyelid	NA	NP	Sclerodermiform BCC (1)	Surgery + RT
Baş Z et al. (2022) [231]	Prevalence of and associated factors for eyelid cancer in the American Academy of Ophthalmology intelligent research in sight registry	Retrospective study	Ophthalmology	49.730 (among 56.610.374 cases)	M (22.898) F (26.832)	Range <20–>65 (NA/NA)	NA	Eyelid	Eyelid	NA	NP	BCC (49.730)	NA
Banerjee P et al. (2022) [232]	The spectrum and clinicopathological correlation of eyelid lesions: twenty years’ experience at a tertiary eye care center in South India	Retrospective study	Oculoplastic surgery	39 (among 992 cases)	NA	Range 21–>60 (NA/NA)	NA	Eyelid	Eyelid	NA	NP	BCC (39)	Surgery
Balchev G et al. (2022) [233]	Glabellar flap technique in oculoplastic surgery	Case series	Ophthalmology	13	NA	Range 49–72 (NA/67.2)	NA	Eyelid	Eyelid	NA	NP	BCC (13)	Surgery
Luo Y et al. (2022) [234]	Deep learning-based fully automated differential diagnosis of eyelid basal cell and sebaceous carcinoma using whole slide images	Retrospective study	Ophthalmology	116 (among 245 cases)	NA	NA	NA	Eyelid	Eyelid	NA	NP	BCC (116)	Surgery
Karlsdóttir SB et al. (2022) [235]	Periocular basal cell carcinoma results and surgical outcome during a 5-year period in a larger Danish population	Case series	Ophthalmology	239 (among 242 cases)	NA	Range NA (70/NA)	NA	Upper eyelid Lower eyelid Medial canthus Lateral canthus	Eyelid Periocular area	NA	NP	BCC (239)	Surgery
Villani A et al. (2022) [236]	The effectiveness of Vismodegib in patients with advanced periocular basal cell carcinoma: a case series of 13 patients	Letter	Dermatology	11 (among 13 cases)	NA	Range NA (76.5/NA)	NA	Upper eyelid Lower eyelid Lateral canthus	Eyelid Periocular area	NA	NP	BCC (11)	Vismodegib + Topical treatments or Surgery or RT
Küronya Z et al. (2022) [237]	Atezolizumab for the treatment of advanced recurrent basal cell carcinoma and urothelial carcinoma of bladder: a case report	Case reports	Urologist	1	F (1)	72	Urothelial carcinoma (1)	Medial canthus	Periocular area	Ulcerated lesion	NP	Follicular BCC (1)	Surgery + RT + Vismodegib + Atezolizumab
Unsworth SP et al. (2022) [238]	Analysis of residual disease in periocular basal cell carcinoma following hedgehog pathway inhibition: follow up to the VISORB trial	Clinical trial	Ophthalmology	32 (among 34 cases)	NA	Range 48–95 (NA/68.5)	NA	Lower eyelid Medial canthus Lateral canthus	Eyelid Periocular area	NA	NP	BCC (32)	Vismodegib ± Surgery
Singh M et al. (2022) [239]	Long-term efficacy and safety of imiquimod 5% and fluorouracil 1% creams in medical monotherapy of complex eyelid basal cell carcinomas	Retrospective study	Ophthalmology	27	NA	Range NA (70.5/NA)	Coronary disease (19); Diabetes mellitus (12); Hypertension (6); Asthma (3); Pulmonary diseases (2)	Eyelid Medial canthus Lateral canthus	Eyelid Periocular area	Pigmented lesion	NP	Noduloulcerative BCC (16) Superficial BCC (11)	Surgery + Fluorouracil 1% or Imiquimod 5%
Chan R et al. (2023) [240]	Mohs surgery for periocular basal cell carcinoma without a Mohs surgeon: the first series in Hong Kong	Case series	Ophthalmology	20	M (10) F (10)	Range 55–91 (78.5/NA)	Pemphigoid (1); BCC (1)	Upper eyelid Lower eyelid Medial canthus	Eyelid Periocular area	Ulcerated nodule	NP	BCC (20)	Surgery
Lin Y et al. (2023) [241]	The clinicopathological analysis of ocular and orbit tumors in southeast of China	Retrospective study	Ophthalmology	59 (among 3.468 cases)	M (29) F (30)	Range NA (54/NA)	NA	Eyelid Conjunctiva Cornea	Eyelid Ocular area	NA	NP	BCC (59)	Surgery
Bagheri A et al. (2023) [242]	A survey on orbital space-occupying lesions during a twelve-year period from a referral center in Iran	Retrospective study	Ophthalmology	11 (among 375 cases)	NA	Range 19–>60 (NA/NA)	NA	Eyelid	Eyelid	NA	NP	BCC (11)	Surgery

**Table 2 diagnostics-15-01244-t002:** Characteristics of the 236 studies included.

Study Type	Number of Articles (%)
Clinical trial	4 (1.7)
Prospective study	11 (4.7)
Prospective case series	10 (4.2)
Case series (>3 patients)	66 (28.0)
Case series with a review of the literature	1 (0.4)
Retrospective study	77 (32.6)
Comparative study	2 (0.8)
Observational study	2 (0.8)
Case reports (max 3 patients)	48 (20.3)
Case report with a review of the literature	6 (2.5)
Letter	9 (3.8)
**Specialization**	**Number of articles (%)**
Ophthalmology	159 (67.5)
Dermatology	27 (11.1)
Plastic surgery	13 (5.6)
Oculoplastic surgery	10 (4.3)
Pathology	7 (3)
Maxillofacial	4 (1.7)
Otorhinolaryngology	3 (1.3)
Surgery	3 (1.3)
Radiotherapy	3 (1.3)
Oncology	2 (0.9)
Radiology	1 (0.4)
Dentistry	1 (0.4)
Urologist	1 (0.4)
Parasitology	1 (0.4)
Electrical engineering	1 (0.4)

**Table 3 diagnostics-15-01244-t003:** Characteristics of the 71.730 patients.

Characteristics	Data
Female	29.782 patients (53.4%)
Male	25.956 patients (46.6%)
Gender not reported	91 studies (38.9%)
Age range	9–105 years
Range not reported	36 studies (15.4%)

**Table 4 diagnostics-15-01244-t004:** Different anatomic localization described in the 236 articles included.

Anatomic Area	Number of Articles (%)
Eyelid and periocular area	115 (48.7)
Eyelid	70 (29.9)
Periocular area	20 (8.5)
Ocular area	9 (3.8)
Caruncle	10 (3.8)
Eyelid, ocular area, and periocular area	5 (2.1)
Eyelid and ocular area	3 (1.3)
Ocular area and periocular area	2 (0.8)
Caruncle and eyelid	1 (0.4)

**Table 5 diagnostics-15-01244-t005:** Assessment performed in the 236 studies included.

Examination	Number of Articles (%)
Clinic	68 (28.8)
Dermoscopy	3 (1.3)
Reflectance confocal microscopy	1 (0.4)
Histopathology	234 (99.2)
Histological subtypes specified	96 (40.7)

**Table 6 diagnostics-15-01244-t006:** Treatment selected in the 236 studies included.

Treatment	Number of Articles (%)
Surgery	154 (65.3)
Surgery plus other treatments	64 (27.1)
Non-surgical therapies	16 (6.8)
Treatment not reported	2 (0.8)

**Table 7 diagnostics-15-01244-t007:** Meta-analysis of the aggregated proportion of included studies for the prevalence of BCC.

Type of Studies	Sample Size	Number of Events	Proportion (%)	95% CI
Clinical trial	339	335	97.6	91.6 to 99.9
Prospective study/Prospective case series	5313	4099	93.0	79.2 to 99.6
Retrospective study	33,882	9302	61.1	48.8 to 72.8
Case reports/Case report with review of the literature	69	66	85.3	77.7 to 91.0
Case series	27,597	6160	73.3	62.1 to 83.3
Letter	1643	1640	89.9	72.8 to 99.0

## Data Availability

The data that support the findings of this study are available from the corresponding author upon reasonable request.

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
