# Peer review of "A Systematic Review and Meta-Analysis of Ocular and Periocular Basal Cell Carcinoma with First-Time Description of Dermoscopic and Reflectance Confocal Microscopy Features of Caruncle Basal Cell Carcinoma"

_diagnostics, 2025, doi:10.3390/diagnostics15101244_

Round 1
Reviewer 1 Report
Comments and Suggestions for Authors
The authors made the largest and most up-to-date systematic review on ocular and periocular basal cell carcinoma, to summarize the clinic characters of this disease. A total of 236 studies were included in this research, though regular results were obtained in this study, the authors still show the message concluded from huge exist papers, which is meaningful.
Author Response
Thank you kindly for the recognition of our work.
Reviewer 2 Report
Comments and Suggestions for Authors
This article presents a good summary of eyelid BCC in literature. Of particular note is that it reports a case of which confocal microscopy was used preoperatively to diagnose the BCC on the caruncle. In general the paper is well written in English, well presented with great detail, and would be a good resource for future studies.
A few minor edits are as follows:
- - Line 128: the “e” is “and” in Italian but it is unclear what is meant by “one article was selected from the references.”
- - 301: needs space between “in” and “14”
- - 316: replace “search in” with “associated with”
- - Reference 270 needs the publication year
Author Response
Thank you very much for your comments.
Comment: Line 128: the “e” is “and” in Italian but it is unclear what is meant by “one article was selected from the references.” Reply: One article was selected after screening of all the references of included studies. Following your suggestion we modified the text and we hope that now it is more clear.
Comment: 301: needs space between “in” and “14”. Reply: Thank you. We corrected the typing error.
Comment: 316: replace “search in” with “associated with”. Reply: Thank you. We modified the text as suggested.
Comment: Reference 270 needs the publication year. Reply: Thank you. We added the publication year.
Reviewer 3 Report
Comments and Suggestions for Authors
Interesting study, just 2 comments: I do not think that you need to cite every accepted and not accepted study- you could sort them out in groups and explain why you accepted the group or not. This would shorten the manuscript which is too long now.
In my opinion, Basal Cell Carcinoma belong to a special group because they grow invasively, but seldom metastasize. Please discuss this in the discussion
Author Response
Thank you for your comments and suggestions.
Comment: Interesting study, just 2 comments: I do not think that you need to cite every accepted and not accepted study- you could sort them out in groups and explain why you accepted the group or not. This would shorten the manuscript which is too long now. Repley: Due to the nature of the manuscript we were obliged to cite every included study. Excluded study were reported only in a supplementary table (not in the biography).
Comment: In my opinion, Basal Cell Carcinoma belong to a special group because they grow invasively, but seldom metastasize. Please discuss this in the discussion. Repley: We modified the text as suggested.